# Eyes on the Road, Words in the Changing Skies: Vision-Language Assistance for Autonomous Driving in Transitional Weather

**Kondapally Madhavi**                                            *cs21resch15001@iith.ac.in*
*Department of Computer Science and Engineering*
*Indian Institute of Technology Hyderabad*

**K Naveen Kumar**                                       *naveenkumar.cse@alumni.iith.ac.in*
*Department of Computer Science and Engineering*
*Indian Institute of Technology Hyderabad*

**C Krishna Mohan**                                                   *ckm@cse.iith.ac.in*
*Department of Computer Science and Engineering*
*Indian Institute of Technology Hyderabad*

**Reviewed on OpenReview:** *https://openreview.net/forum?id=PCEDvdVJon*

## Abstract

The rapid advancement of autonomous vehicle technology (AVT) necessitates robust scene perception and interactive decision-making, particularly under adverse weather conditions. While significant progress has been made in extreme weather scenarios like cloudy, foggy, rainy, and snowy, a critical challenge remains in transitional weather conditions, such as the shift from cloudy to rainy, foggy to sunny, etc. These dynamic environmental changes degrade the performance of conventional vision-language systems by causing unpredictable illumination changes and partial occlusions, which are inadequately represented in current AVT datasets. This lack of continuous, transitional training data compromises model robustness and ultimately affects safety and reliability. On the other hand, Vision-language Models (VLMs) enable interpretable reasoning in autonomous driving through tasks such as image captioning and visual question answering. However, current VLMs, designed for clear weather, perform poorly in transitional conditions and rely on computationally expensive LLMs. This leads to high memory usage and slow inference, which is unsuitable for real-time decision making in AVT. To address these limitations, we propose Vision-language Assistance for Autonomous Driving under Transitional Weather (VLAAD-TW), a lightweight framework with a novel cross-modal spatiotemporal reasoning architecture that robustly interprets and acts on multimodal data. The VLAAD-TW framework integrates a Feature Encoder for Transitional Weather (FETW), a lightweight backbone for robust visual feature extraction, with a Spatiotemporal Contextual Aggregator (SCA), which models dynamic weather-induced changes. It uses a Selective Attention-guided Fusion Module (SAFM) to balance visual and linguistic cues for a unified representation dynamically. Finally, a Semantic Text Generator (STG) fuses these representations to produce context-aware driving information, adapting in real time to both current and predicted weather states. Further, we introduce AIWD16-text dataset, an adverse intermediate weather driving dataset for vision language tasks, which features sixteen transitional weather states created using a Stochastic Conditional Variational Autoencoder (SC-VAE) and enriched with manual annotations of image captions and open-ended question-answer pairs. An extensive evaluation of the AIWD16-text and DriveLM datasets demonstrates VLAAD-TW's high performance in BLEU and ROUGE scores, with low memory and computational requirements, confirming its effectiveness in challenging weather conditions.

# 1 Introduction

Autonomous vehicle technology (AVT) has made substantial progress in advanced driver assistance systems for real-world applications (Li et al., 2024b; Hwang et al., 2025). However, the functionality of AVT is significantly challenged by adverse weather conditions, such as rain, fog, and snow (Wang et al., 2024; Lu et al., 2025). Also, these weather scenarios drastically degrade the quality of visual input from camera sensors, posing difficulties for vision-language models (VLM) to generate accurate and reliable descriptions of driving scenes (Xie et al., 2024; Luo et al., 2025). Moreover, sudden weather changes, referred to as transitional weather conditions like cloudy to rainy or rainy to foggy, obstruct the recognition of vehicles, pedestrians, and traffic signs due to variations in weather intensity (Sun et al., 2022).

**Need for the study of transitional weather and its dataset.** Transitional weather refers to intermediate weather conditions that occur when weather changes from one type to another. It involves gradual shifts in weather conditions. Examples include transitions such as foggy weather turning into rainy or rainy shifting to cloudy, with variations in the weather intensity (Sun et al., 2022). Studying transitional weather conditions is critical in real-world environments, as abrupt transitions between weather conditions are common and pose challenges to AVT. Existing datasets (Caesar et al., 2019; Sakaridis et al., 2018) primarily focus on either clear weather or extreme conditions, leaving a notable gap in understanding and modeling transitional weather scenarios. To address this limitation, we introduce the AIWD16-text dataset, designed for vision-language assistance in AVT. The dataset encompasses sixteen transitional weather scenarios, including cloudy to rainy (CR), sunny to foggy (SF), sunny to rainy (SR), cloudy to snowy (CSn), cloudy to foggy (CF), snowy to rainy (SnR), snowy to foggy (SnF), foggy to rainy (FR) and vice versa. Additionally, we provide manually annotated, generated images to facilitate the creation of open-ended question–answer pairs, enabling richer vision-language modeling under challenging weather transitions. To our knowledge, this is the first dataset tailored for image captioning and visual question-answering (VQA) tasks for AVT under transitional weather conditions.

**Transitional weather impact on VLMs.** Vision-language models (VLMs) offer a promising avenue for interpretable reasoning in autonomous driving tasks to interpret their surroundings semantically (Ma et al., 2024b; Moeller et al., 2025). They enable tasks like image captioning (e.g., "A pedestrian is crossing the street") and visual question answering (e.g., "Is the traffic light red?"), which is crucial for enhancing situational awareness and providing an interpretable basis for real-time decision-making in autonomous driving systems (Kuchibhotla et al., 2025; Zhang et al., 2024a). However, existing VLMs (Chen et al., 2024b) are typically designed for static, clear weather and often fail to perform reliably in dynamic transitional weather scenarios due to high visual variability caused by changing weather patterns and reduced illumination, making it challenging to detect and identify crucial road elements, such as pedestrians and vehicles. Moreover, a key barrier to their adoption in AVT is their reliance on computationally expensive Large Language Models (LLMs), which leads to high memory demands and slow inference times, rendering them unsuitable for the real-time decision-making required for autonomous driving (Gopalkrishnan et al., 2024).

**Limitations of existing methods.** Table 1 presents a summary of existing VLMs, highlighting the focus of our work on transitional weather conditions. While current VLM research on image captioning and VQA (Li et al., 2022; Byun et al., 2024; Li et al., 2023) focuses predominantly on general-purpose applications, these models lack exposure to driving-specific scenarios, limiting their ability to understand crucial road elements and safety-critical situations. Similarly, existing VQA methods for AVT are primarily limited to ideal daylight conditions, failing to address real-world complexities such as transitional weather conditions (Gopalkrishnan et al., 2024; Park et al., 2024). The limited availability of training data representing these diverse weather scenarios further compounds the performance constraints. Although few VLMs target specialized tasks like traffic rule extraction (Li et al., 2024a) and action explanation (Ma et al., 2024b), their reliance on curated datasets and controlled environments constrains their effectiveness in handling unpredictable real-world driving conditions (Kuchibhotla et al., 2025; Wu et al., 2025; Zang et al., 2025). Moreover, these approaches typically depend on large-scale models (Li et al., 2022; Touvron et al.) with over a billion parameters, requiring expensive hardware and suffering from slower inference speeds, making them impractical for real-time autonomous driving. Overall, existing methods suffer from three key limitations, *(i)* insufficient datasets

Table 1: Comparison of existing vision language methods with the proposed method. R→Rainy, C→Cloudy, S→Sunny, F→Foggy, Sn→Snowy, ✓→Present, ✗→Not present, '-'→Not applicable.

| Method | Task | Autonomous driving scenarios | Data generation | Adverse weather conditions | Transition, transition classes, and real data |
|---|---|---|---|---|---|
| CLIP (Radford et al., 2021) | Image Captioning | ✗ | ✗ | - | ✗, -, - |
| Concept-gridlock (Echterhoff et al., 2024) | | ✓ | ✗ | - | ✗, -, - |
| IL-CLIP (Zheng et al., 2024) | | ✗ | ✗ | - | ✗, -, - |
| MAFA (Byun et al., 2024) | | ✗ | ✗ | - | ✗, -, - |
| VeCLIP (Lai et al., 2024) | | ✗ | ✓ | - | ✗, -, - |
| ExCLIP (Moeller et al., 2025) | | ✗ | ✗ | - | ✗, -, - |
| SOLO (Chen et al., 2024b) | | ✗ | ✗ | - | ✗, -, - |
| LingoQA (Marcu et al., 2024) | VQA | ✓ | ✓ | - | ✗, -, - |
| EM-VLM4AD (Gopalkrishnan et al., 2024) | | ✓ | ✗ | C, R, Sn | ✗, -, - |
| VLAAD (Park et al., 2024) | | ✓ | ✓ | - | ✗, -, - |
| CocoCon (Maharana et al., 2024) | | ✗ | ✗ | - | ✗, -, - |
| CODA-VLM (Chen et al., 2025) | Traffic rule extraction | ✓ | ✓ | - | ✗, -, - |
| LLaDA (Li et al., 2024a) | | ✓ | ✗ | - | ✗, -, - |
| LaMPilot (Ma et al., 2024b) | Driving actions and explanations | ✓ | ✓ | - | ✗, -, - |
| SGDCL (Cao et al., 2024) | | ✓ | ✗ | - | ✗, -, - |
| BLIP (Li et al., 2022) | Image Captioning + VQA | ✗ | ✓ | - | ✗, -, - |
| BLIP-2 (Li et al., 2023) | | ✗ | ✓ | - | ✗, -, - |
| **VLAAD-TW (Ours)** | | ✓ | ✓ | **S, R, C, F, Sn** | ✓, 16, ✓ |

for transitional weather conditions, *(ii)* limited output interpretability, and *(iii)* deployment constraints of large-scale models in real-time systems.

To address the above challenges, we propose VLAAD-TW, a lightweight vision-language assistance framework specifically designed for autonomous driving under transitional weather conditions. The overview of the VLAAD-TW framework, illustrated in Figure 1, is designed for image captioning and VQA. VLAAD-TW introduces a novel cross-modal spatiotemporal reasoning architecture that robustly interprets and acts on multimodal data in transitional weather conditions. The framework integrates *(a)* a Feature Encoder for Transitional Weather (FETW), a lightweight residual backbone fine-tuned to maintain feature stability across varying intensities of clouds, rain, fog, and snow, *(b)* a Spatiotemporal Contextual Aggregator (SCA), that jointly captures spatial scene structure and temporal evolution to model dynamic weather-induced changes, *(c)* a Selective Attention-guided Fusion Module (SAFM) that dynamically balances visual cues (e.g. obstacles) and linguistic descriptions (e.g., 'heavy rain,' 'low visibility') to form a unified safety-critical representation, and *(d)* a Semantic Text Generator (STG) that fuses these representations to produce context-aware driving information, adapting in real time transitional weather states. Additionally, we introduce AIWD16-text, a novel vision language dataset tailored for transitional weather scenarios, created using a stochastic conditional variational autoencoder (SC-VAE) to enhance model diversity and generalization in adverse and transitional conditions. This dataset is manually annotated to support tasks such as image captioning and VQA, offering a valuable resource for advancing research in this domain. To the best of our knowledge, this work represents the first comprehensive effort to address vision-language tasks in transitional weather conditions for AVT. The main contributions of our work are as follows.

1. **A novel, end-to-end framework for Vision-Language Assistance in Autonomous Driving (VLAAD-TW).** We introduce a first-of-its-kind model specifically developed to perform complex VQA and image captioning tasks under challenging transitional weather conditions, addressing a critical and previously underexplored challenge in autonomous vehicle perception.

2. **A lightweight and robust architectural design for transitional weather perception.** Our VLAAD-TW model incorporates a specialized Feature Encoder for Transitional Weather (FETW) and a Spatiotemporal Contextual Aggregator (SCA). This design ensures both computational efficiency for real-time applications and robust feature stability, mitigating the adverse effects of transitional weather phenomena like cloudy to rainy and rainy to foggy, etc., on perception.

3. **AIWD16-text dataset.** We introduce AIWD16-text, a novel vision-language dataset generated using a stochastic conditional variational autoencoder (SC-VAE) to address the challenges of transitional weather

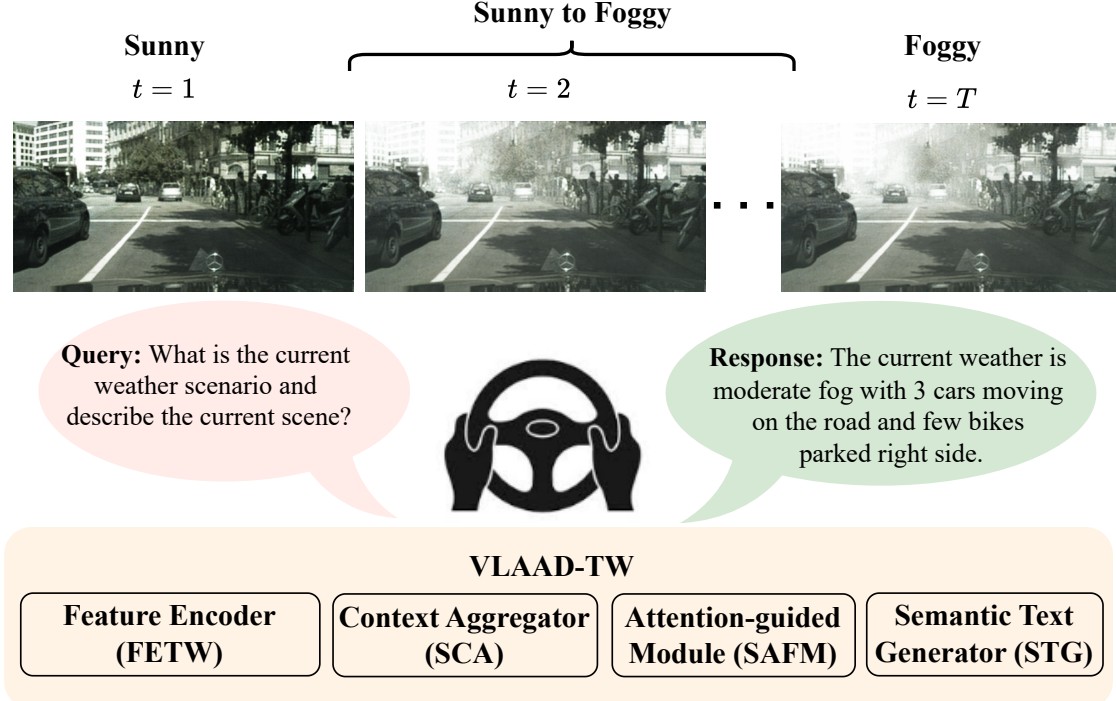

Figure 1: Overview of the proposed VLAAD-TW approach: Transitional weather images generated by the SC-VAE are input to VLAAD-TW, which generates captions and answers to provide vision-language assistance.

conditions in autonomous driving. This dataset, which includes a total of sixteen distinct weather states, features a rich combination of manually curated image captions and open-ended visual question-answer pairs, making it an ideal resource for training and evaluating vision-language models in this domain.

4. **Comprehensive evaluation.** We present a comprehensive evaluation of our VLAAD-TW framework, benchmarking its performance against several state-of-the-art vision-language models. This evaluation is conducted on both our new AIWD16-text dataset and the well-established DriveLM dataset. Furthermore, we provide a detailed analysis of VLAAD-TW's computational complexity, demonstrating its efficiency and real-world applicability compared to existing, larger state-of-the-art models.

**Paper outline:** Section 2 reviews prior work on synthetic weather generation, vision-language models (VLMs) for image captioning and visual question answering (VQA), and the application of VLMs in AVT. Section 3 discusses essential background information for weather transition generation. Section 4 illustrates the details of our proposed method, which comprises two stages, transitional weather data generation and vision language modeling for autonomous driving. Section 5 presents a comprehensive assessment of our proposed method for dataset generation and vision language modeling tasks. Section 6 provides a detailed ablation study of our work. Section 7 provides limitations of our work and future research directions. Finally, Section 8 summarizes key research findings.

## 2 Related Work

This section discusses recent works on vision language models for image captioning, visual question and answer (VQA) and other notable works.

## 2.1 Synthetic Weather Generation

To address the challenges of limited data availability for AVT in inclement weather, numerous studies (Diaz et al., 2022; Li et al., 2025; Marathe et al., 2023) have presented methodologies for generating synthetic data in autonomous vehicles. Park et al. (2024) introduced a multi-modal instruction tuning dataset to facilitate language models in learning instructions across driving scenarios. Marcu et al. (2024) introduced the LingoQA benchmark for visual question answering in autonomous driving. Li et al. (2025) proposed a spatial-temporal consistent diffusion framework, DrivingDiffusion, to generate realistic multi-view videos for autonomous driving. Marathe et al. (2023) introduced WEDGE, a synthetic dataset generated with a vision-language generative model via prompting. El-Shair et al. (2024) introduced a large-scale stereo image dataset that captures a wide spectrum of challenging real-world environmental scenarios. Mușat et al. (2021) generated seven diverse weather conditions using GAN and CycleGAN. Hu et al. (2021b) synthesized rainy images to create a new rainy dataset called rainy cityscapes for single-image rain removal. Most of these methods generate discrete weather conditions. Unlike these, Sun et al. (2022) introduced a synthetic driving dataset known as SHIFT for performing scene perception tasks under discrete and continuous domain shifts using a Carla simulator (Dosovitskiy et al., 2017). However, compared to generative-based approaches, simulators often fall short in variability, realism, complexity, and scalability. *To overcome these limitations, we generate the transitional weather data using generative approaches that allow us to vary weather intensities and annotate them to create a rich set of open-ended question–answer pairs for driving scenarios.*

## 2.2 VLMs for Image Captioning and VQA

Vision-language models connect the capabilities of language and vision models by integrating images and text into a shared latent representation. They utilize cross-modal pre-training tasks to establish relationships between visual and textual data, inspiring the creation of numerous models in multimodal learning (Gopalkrishnan et al., 2024). Li et al. (2022) proposed BLIP, a unified framework for vision-language understanding and generation by bootstrapping pre-training with noisy web data. Li et al. (2023) proposed BLIP-2 that improves upon BLIP by utilizing frozen image encoders and large language models for vision-language pre-training. Zheng et al. (2024) design an iterated learning algorithm that improves compositionality in large vision-language models. Radford et al. (2021) proposed a pre-training task that pairs text captions with images, resulting in CLIP, which learns advanced image representations and demonstrates exceptional zero-shot transfer performance across various image classification tasks. Byun et al. (2024) introduced MAFA, specifically designed to tackle false negatives in VLP. Ma et al. (2024a) introduced a vision-language instruction tuning (VLIT) approach using contrastive learning (C3L), featuring a content relevance module to align VLIT data with images and a contrastive learning module to enhance data generation. Shukor et al. (2022) proposed ViCHA, incorporating three key components: hierarchical cross-modal contrastive alignment, self-supervised masked image modeling, and supervision using CLIP-derived Visual Concepts. Li & Jiang (2024) proposed a method that leverages cross-modal pre-training and semantic modeling to enhance video anomaly detection, achieving improved interpretability and performance. Moeller et al. (2025) introduced a second-order attribution method for dual-encoder models, showing that when applied to CLIP, it effectively reveals fine-grained correspondences between caption tokens and image regions. *Although the proposed methods are efficient and focus predominantly on general-purpose applications, these models lack exposure to driving-specific scenarios, which limits their ability to understand critical road elements and safety-related situations.*

## 2.3 VLMs for Autonomous Driving

Autonomous driving systems predominantly rely on visual features, but incorporating linguistic features can improve their interpretability and aid in identifying novel traffic scenarios. This advantage has fueled growing research into leveraging multimodal data to train language models as autonomous driving agents (Ma et al., 2024b). Gopalkrishnan et al. (2024) proposed EM-VLM4AD, a multi-frame vision-language model optimized for efficient visual question answering in autonomous driving, focusing on memory efficiency and reduced computational demands. Park et al. (2024) introduced VLAAD, a multimodal LLM driving assistant designed to generate detailed captions for autonomous driving scenarios. Pan et al. (2024) presented

Table 2: Summary of adopted notations

| Notation | Definition |
|---|---|
| $\mathcal{D}$ | Training set |
| $\mathcal{M}$ | Size of training set |
| $x_i, y_i$ | Input image pair (Ex. $x_i$: Sunny image and $y_i$: Rainy image) |
| $l_i$ | One-hot encoded labeled vector |
| $\Psi$ | Variational parameters |
| $z$ | Latent variable |
| $u$ | Input features |
| $\Theta$ | Model parameters |
| $\mu$ | Mean of gaussian distribution |
| $\sigma$ | Variance of gaussian distribution |
| $\beta$ | Coefficient |
| $\mathcal{L}$ | Loss |
| $t$ | Transition sequence |
| $T$ | Length of transition sequence |
| $\beta_0$ & $\beta_1$ | Coefficients |
| $\epsilon$ | Error |
| $\hat{u}$ | Reconstructed input |
| $\mathcal{I}$ | Covariance |
| $\mathcal{V}$ | Feature vectors obtained from CNN |
| $\mathcal{P}$ | Processed feature vectors |
| $\alpha$ | Attention vector |
| $\mathcal{C}$ | Combined context vector |
| $h$ | Hidden state |

a vision-language planning framework to bridge the gap between linguistic understanding and autonomous driving. Chen et al. (2024a) developed an architecture that combines vectorized numeric modalities with a pre-trained LLaMA-7b Touvron et al. to tackle driving question-answering tasks. Their two-step training approach involves grounding vector representations into interpretable embeddings for the frozen LLaMA model, then fine-tuning the LLM using LoRA (Hu et al., 2021a). Similarly, DriveGPT4 Xu et al. (2024) utilizes LLaMA as its backbone language model and CLIP as a visual encoder, processing traffic scene videos and prompt text to generate responses and low-level vehicle control signals. In contrast, Li et al. (2024a) proposed LLaDA, a vision-language framework capable of interpreting traffic rules in new locations with zero-shot generalizability. Ma et al. (2024b) introduced a framework integrating LLMs into autonomous driving systems, enhancing their ability to interpret and follow user commands. Additionally, Cao et al. (2024) proposed the Semantic-Guided Dynamic Correlation Learning (SGDCL) framework, leveraging semantic-guided and dynamic correlation modules to enhance explainable autonomous driving. However, these methods remain largely limited to ideal daylight conditions, failing to address real-world complexities such as transitional weather. *Hence, in our work, we propose a lightweight and efficient vision-language assistant capable of generating captions and responding to queries in autonomous driving scenarios under transitional weather conditions.*

## 3   Preliminaries

This section provides the essential background information required to comprehend the remainder of the paper. A summary of the notations used throughout this work is presented in Table 2.

### 3.1   Latent Space Representation

The latent space of a Conditional Variational Autoencoder (C-VAE) is a hidden, low-dimensional representation of the input data. Unlike a traditional autoencoder, which maps an input to a single point, the C-VAE learns a probabilistic distribution, specifically, a Gaussian for each input. This approach prevents overfitting and ensures that the latent space is continuous and well-structured, a property that is essential

for a generative model. The inherent smoothness of this latent space is a key feature of the C-VAE, as it enables seamless and meaningful interpolation between distinct data points. By generating a new point in the latent space and decoding it, the model can synthesize novel data that blends the characteristics of the original inputs. This property is crucial for our work, allowing us to generate the progressive weather transitions required for our dataset.

### 3.2 Data Interpolation

To generate a diverse and continuous spectrum of transitional weather states for the proposed AIWD16-text dataset, we leverage the power of latent space interpolation within C-VAE. The process begins by encoding two distinct input images captured under different weather conditions to obtain their corresponding latent vectors, $z_x$ and $z_y$, and their one-hot encoded condition vectors, $l^x$ and $l^y$. A series of intermediate latent vectors, $z$, and blended condition vectors, $l$, are then generated via linear interpolation, The C-VAE's decoder then utilizes these interpolated latent and condition vectors to transform them into a progressive sequence of images. This technique allows for synthesising a smooth and realistic evolution of visual features between distinct weather conditions, with $\beta$ providing fine-grained control over the blend of characteristics.

### 3.3 Transitional Weather Conditions

Weather transitions are common in the real world and pose a challenge for AVT, particularly in rain, clouds, and fog. Existing datasets typically capture stable weather conditions (e.g., cloudy or rainy), but real-world transitions, such as from sunny to rainy, involve varying intensities. For example, transitioning from sunny to rainy involves intensities ranging from light to heavy rain (Sun et al., 2022). Recent studies highlight challenges posed by continuous weather shifts for learning systems, underscoring the need for datasets that capture dynamic weather changes (Sun et al., 2022). Our work provides such a dataset using customized conditional variational autoencoder, enabling vision language modeling for AVT development under transitional weather conditions.

**Importance of addressing weather transitions in autonomous vehicles.** Transitional weather conditions significantly impact scene perception performance in autonomous driving. Although existing datasets such as Foggy Cityscapes (Sakaridis et al., 2018) and MultiWeatherCity (Muşat et al., 2021) capture adverse weather scenarios, they mainly focus on discrete weather conditions such as heavy rain or intense fog. On the other hand, real-world driving environments often involve gradual weather transitions (for example, foggy to rainy or rainy to sunny). These transitions introduce additional complexities that go beyond those encountered in extreme weather conditions.

**Unique challenges posed by transitional weather.** Existing AVT datasets and models inadequately address critical challenges associated with weather transitions: *(i)* Unlike static weather states, transitions involve dynamic changes in visibility and illumination, which can disrupt perception models trained on fixed conditions, *(ii)* autonomous systems rely on stable environmental features for perception. However, weather transitions introduce inconsistencies in visual cues, leading to misclassification or unreliable predictions.

**Limitations of existing models in handling transitional weather.** Most VLMs are trained on datasets with extreme or static weather conditions, making them ill-equipped to handle gradual transitions. Their primary limitations include: *(i)* Models trained on clear or extreme weather conditions lack the adaptability to generalize across intermediate weather changes, *(ii)* existing models often rely on static images or short sequences, missing the gradual evolution of visual features over time, *(iii)* during transitions, environmental features may not match any single weather condition category, causing prediction errors. These limitations reduce model robustness in real-world deployments, where smooth weather transitions are common. Addressing these gaps is essential for developing more reliable and adaptable autonomous vehicle perception systems.

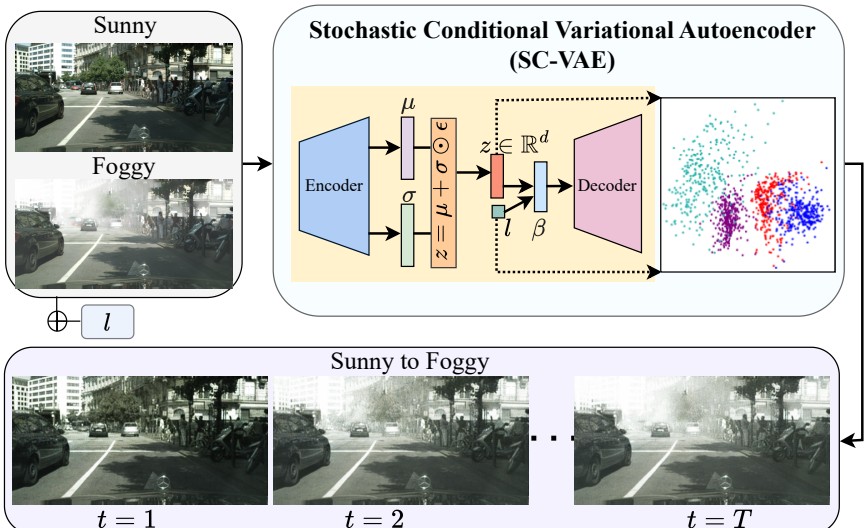

Figure 2: The proposed transitional weather data generation approach using a customized conditional variational autoencoder. Here $y = [0, 1, 0]$ for sunny weather and $[0, 1, 1]$ for foggy weather.

## 4 Proposed Method

**Method overview.** Our proposed method comprises two stages: *(i)* generate a benchmark transitional weather dataset for AVT and *(ii)* perform vision-language tasks such as image captioning and VQA on the generated dataset. Figure 2 presents our proposed transitional weather data generation approach. First, we input driving scene images $x$ and a one-hot encoded vector $l$ representing the corresponding weather condition into the SC-VAE. The SC-VAE's encoder processes both the image features and the condition vector to generate the parameters $(\mu, \log \sigma^2)$ of a latent distribution. To synthesize transitional images, we perform a linear interpolation between two latent embeddings while using a blended condition vector to guide the decoder. This process allows the decoder to produce a continuous and controlled evolution of visual features, resulting in a sequence of transitional weather images with varying intensities. Annotations are then manually created to support both vision-language tasks effectively. Figure 3 illustrates the framework of our proposed VLAAD-TW architecture, which integrates a series of components to effectively generate textual descriptions and answers to questions from images. A sequence of transitional weather images is first input to the Feature Encoder for Transitional Weather (FETW), a lightweight MobileNetV2 backbone (Sandler et al., 2018) that extracts high-level visual features resilient to weather-related degradation. Concurrently, input text is processed by the Spatiotemporal Contextual Aggregator (SCA), which consists of a series of LSTM networks that transform the word sequence into a coherent, context-aware representation. The outputs from both the FETW and SCA modules are then passed to the Selective Attention-Guided Fusion Module (SAFM). This module dynamically weighs the importance of both visual features and the linguistic context, fusing them into a unified, salient representation. Finally, the fused context is fed to the Semantic Text Generator (STG), which consists of LSTM layers that combine the context vector with previous outputs to maintain sequence coherence. A dense layer with softmax activation predicts the next word, generating a refined and context-aware textual output.

### 4.1 Transitional Weather Data Generation

Let $\mathcal{D} = \{(x_i, y_i, l_i^x, l_i^y)\}_{i=1}^{\mathcal{M}}$ denote the training dataset, where each pair of images $(x_i, y_i)$ is captured under two distinct stable weather conditions. The corresponding one-hot encoded labels for these conditions are denoted by $l_i^x$ and $l_i^y$, respectively, and $\mathcal{M}$ is the total number of training samples. Our stochastic conditional variational autoencoder (SC-VAE) consists of an encoder, denoted as $q_\Psi(z \mid x, l)$, which processes each image and its condition vector independently. For a given input pair $(x_i, y_i)$, the encoder is applied to each image

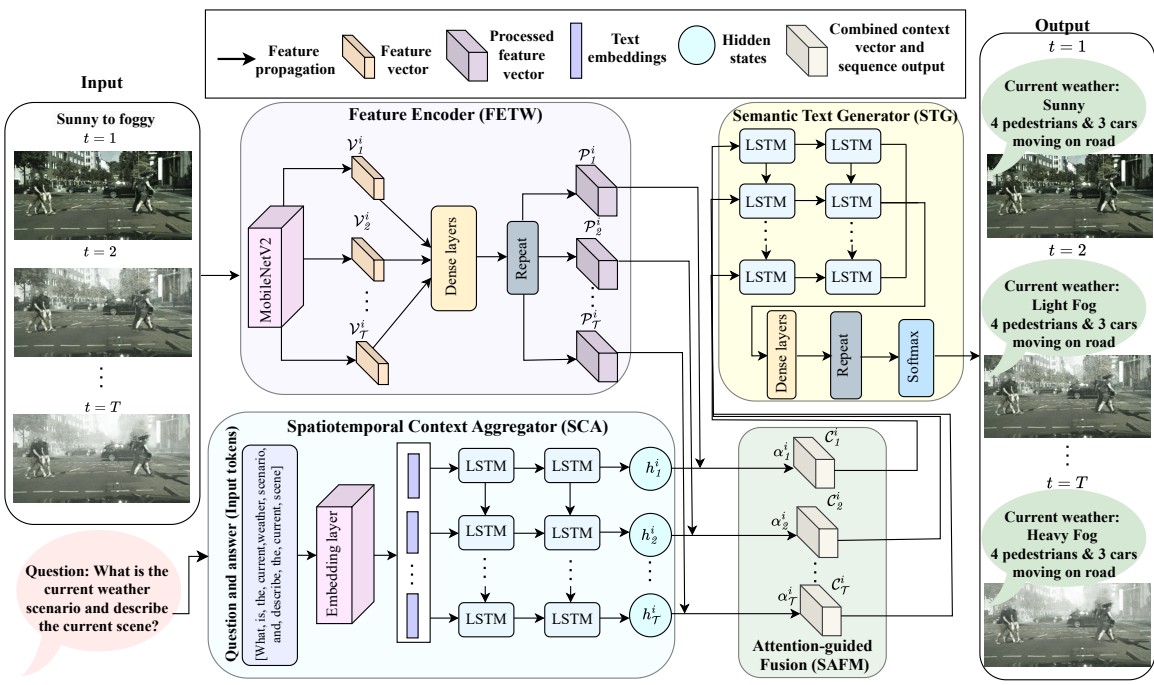

Figure 3: The proposed architecture of the VLAAD-TW. We input transitional weather sequences (e.g. Sunny to Foggy) to the Feature encoder (FETW) having MobileNetV2 backbone for feature extraction, refined by dense layers, and contextualized using context generator (SCA) and attention-guided fusion module. The semantic text generator (STG) integrates the context vector with LSTM outputs, predicting the next word via a dense layer with softmax activation.

to generate two distinct latent representations, $z_x$ and $z_y$. The latent variables for each are generated from Gaussian distributions specified by the corresponding mean and variance values,

$$(\mu_x, \log(\sigma_x^2)) = \text{Encoder}_\Psi(x_i, l_i^x), \tag{1}$$

$$z_x \sim \mathcal{N}(\mu_x, \sigma_x^2), \tag{2}$$

$$(\mu_y, \log(\sigma_y^2)) = \text{Encoder}_\Psi(y_i, l_i^y), \tag{3}$$

$$z_y \sim \mathcal{N}(\mu_y, \sigma_y^2). \tag{4}$$

Where $\Psi$ represents the encoder's variational parameters, this process generates a pair of latent representations $(z_x, z_y)$, which are then used to perform a controlled interpolation to synthesize transitional weather images. The SC-VAE facilitates a stochastic mapping from the input images to their latent representations. The decoder, acting as a generative model, utilizes an interpolated latent representation $z$ and a blended condition vector $l$ to synthesize new images. Specifically, for a given pair of latent representations $(z_x, z_y)$ and their corresponding condition vectors $(l^x, l^y)$, the interpolated inputs are calculated as follows,

$$z = \beta z_x + (1 - \beta)z_y, \tag{5}$$

$$l = \beta l^x + (1 - \beta)l^y, \tag{6}$$

where $\beta \in [0, 1]$. The decoder then uses these interpolated inputs to generate a new image:

$$\hat{x} = \text{Decoder}_\Phi(z, l). \tag{7}$$

Figure 4 illustrates the fundamental process of latent data interpolation, which is central to creating the transitional states of our dataset. This process enables the generation of an interpolated image that smoothly blends the weather characteristics of the input pair $(x_i, y_i)$. The parameter $\beta$ provides fine-grained control

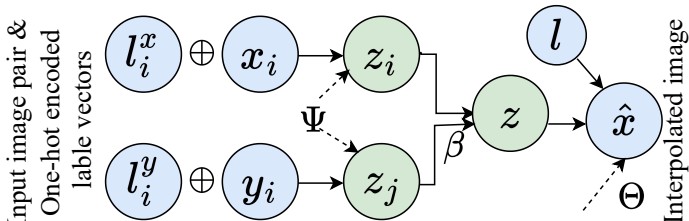

Figure 4: Latent space interpolation using SC-VAE. The encoder takes an input image pair $(x_i, y_i)$ and their corresponding one-hot encoded label vectors $(l_i^x, l_i^y)$ to produce distinct latent representations $(z_i, z_j)$. By linearly interpolating the latent and label vectors, we generate a new blended latent vector and a new blended label vector. The SC-VAE's decoder then uses these to synthesize a novel image that smoothly blends the characteristics of the original two conditions.

over the relative intensities of the two weather conditions, which is a key advantage of our conditional approach. The joint distribution learned by the SC-VAE is now conditional on the input data and its associated label. The joint distribution is given by $p_\Phi(\hat{x}, z \mid l)$, which is the product of the conditional prior distribution $p(z \mid l)$ and the conditional distribution $p(\hat{x} \mid z, l)$. The encoder learns the approximate posterior distribution $q_\Psi(z \mid x, l)$.

To enable backpropagation through the stochastic sampling process, our SC-VAE employs the reparameterization trick, expressed as

$$z = \mu + \sigma \odot \epsilon, \tag{8}$$

where $\epsilon$ is drawn from a standard normal distribution with mean 0 and identity covariance matrix $\mathcal{I}$.

The objective function is designed to minimize the reconstruction loss and the Kullback-Leibler (KL) divergence. The total loss is given by:

$$\mathcal{L}(x, z, l, \Psi, \Phi) = \mathcal{L}_{recon}(x, \hat{x}) + \mathcal{L}_{KL}(q_\Psi(z \mid x, l) \parallel p(z \mid l)). \tag{9}$$

More specifically, the loss is defined as:

$$\mathcal{L}(x, z, l, \Psi, \Phi) = \parallel x - \hat{x} \parallel^2 + \mathrm{KL}(q_\Psi(z \mid x, l) \parallel p(z \mid l)), \tag{10}$$

where $\hat{x} = \mathrm{Decoder}_\Phi(z, l)$ is the reconstructed input. The KL divergence term encourages the learned latent distribution to align with the conditional prior distribution, ensuring the latent space is well-structured and disentangled based on the weather conditions.

In summary, we address the critical lack of transitional weather data by introducing a novel data generation methodology. We feed image pairs representing distinct weather conditions into our SC-VAE and perform a controlled interpolation in the latent space. This process enables us to synthesize transitional weather images with varying intensities of fog, rain and snow. For example, when inputting sunny and foggy images, the SC-VAE's encoder generates two distinct, low-dimensional latent embeddings. The decoder then utilizes a blended latent embedding and a blended condition vector to generate images showing a smooth, data-driven weather transition between the original conditions. This novel process enables us to create AIWD16-text, a unique dataset that facilitates the training of our model on diverse and transitional weather scenarios. By explicitly isolating and controlling weather effects, this approach leads to more robust image captioning and VQA performance across various weather conditions, which is crucial for autonomous driving applications.

## 4.2 VLAAD-TW

Figure 3 illustrates the VLAAD-TW framework, which integrates a series of components to effectively generate textual descriptions and answers to questions from images. It consists of a Feature Encoder for Transitional Weather (FETW), built on a lightweight MobileNetV2 backbone, that extracts high-level visual

features resilient to weather degradation. A Spatiotemporal Contextual Aggregator (SCA), comprising a series of LSTM networks, processes the input text into a coherent, context-aware representation. A Selective Attention-Guided Fusion Module (SAFM) dynamically weighs and fuses the visual features from the FETW and the linguistic context from the SCA. Finally, a Semantic Text Generator (STG), consisting of LSTM layers, synthesizes the fused information to produce a refined and context-aware textual output.

**Feature Encoder for Transitional Weather (FETW).** Let a weather transition sequence of length $\mathcal{T}$, each image is processed by our FETW. This component, built on a lightweight MobileNetV2 backbone (Sandler et al., 2018), is designed to extract high-level visual features that are robust against the degradation caused by dynamic weather. It outputs a sequence of high-dimensional visual embeddings $\{\mathcal{V}_1, \mathcal{V}_2, \ldots, \mathcal{V}_\mathcal{T}\}$, capturing salient spatial and semantic information from each frame. These features are then fed into a series of dense layers that project them into a lower-dimensional, task-specific representation, aligning their dimensions with the expected input of the Semantic Text Generator (STG). This transformation facilitates a seamless integration with the text generation modules by ensuring each output token is conditionally grounded in a robust visual context. To enhance the model's resilience and prevent overfitting to specific noise patterns, dropout is employed during training, encouraging the model to learn more generalized and diverse feature representations.

**Spatiotemporal Contextual Aggregator (SCA).** This component processes the input text, such as a question or a prompt, to generate a coherent linguistic representation. It first tokenizes the input text into smaller units, with each token mapped to an integer from a predefined vocabulary. These indices are then transformed into dense vectors via an embedding layer, which captures semantic relationships between words. The embedded tokens are subsequently processed by an LSTM network, which is responsible for modeling long-range dependencies and maintaining a robust contextual state across the sequence. For an input text of length $\mathcal{T}$, the LSTM outputs hidden states $\{h_1, h_2, \ldots, h_\mathcal{T}\}$, which serve as context-aware representations for the downstream cross-modal fusion. This process is crucial for providing the linguistic information necessary for the SAFM to interpret the visual scene accurately.

**Selective Attention-Guided Fusion Module (SAFM).** Our attention mechanism, the SAFM, is specifically designed to enhance the interaction between the visual and linguistic modalities. It calculates attention scores $\alpha_1, \alpha_2, \ldots, \alpha_N$ through a dot product between the hidden states of the SCA and the image features extracted by the FETW. These scores dynamically weigh the relevance of specific visual features for each word in the generated sequence, with higher scores indicating a stronger focus on the most important image regions. Using these scores, the module computes a context vector $\mathbf{v}_{\text{context}}$ as a weighted sum of the image features, $\mathbf{v}_{\text{context}} = \sum_{i=1}^{N} \alpha_i \mathcal{V}_i$, where $\mathcal{V}_i$ represents the $i$-th image feature from the FETW. The context vector encapsulates the most relevant and salient image information, effectively guiding the model to generate accurate and contextually appropriate descriptions even in the presence of weather-related visual degradation. This cross-modal fusion is a crucial step that distinguishes our approach from models that process visual and textual information in isolation.

**Semantic Text Generator (STG).** The STG is responsible for synthesizing the final textual output. It processes the context vector $\mathbf{v}_{\text{context}}$, which is the fused output from the SAFM, alongside the output of the LSTM from the SCA. This combined input ensures that the text generation is grounded in both the most relevant visual features and the overall linguistic context. An additional LSTM layer is employed within the STG to process this combined context vector and the previous decoder output. This layer maintains the contextual flow of the generated text, ensuring it remains coherent and semantically relevant to the dynamic visual input. Finally, a dense layer with a softmax activation function is applied to the LSTM's output. This produces a probability distribution over the entire vocabulary, enabling the prediction of the next word in the sequence based on a rich, multimodal understanding of the scene.

**Rationale for VLAAD-TW Design.** The design of our proposed VLAAD-TW framework is not a straightforward combination of existing components but a deliberate selection of architectures tailored to the unique challenges of real-time autonomous driving under transitional weather conditions. We employ a lightweight MobileNetV2 backbone for visual feature extraction within our FETW module. This choice is predicated on its highly efficient architecture, which utilizes depthwise separable convolutions and an inverted residual structure to deliver competitive performance with minimal computational requirements.

Table 3: Overview of the dataset and implementation details for the proposed VLAAD-TW approach. B → Batch size, I → Input size, Op → Optimizer, and lr→Learning rate.

| Task | AIWD16-text Dataset | | | | Data Generation | | | | | Vision Language Assistance (VLAAD-TW) | | | |
|---|---|---|---|---|---|---|---|---|---|---|---|---|---|
| | No of transition states | Sequence length (T) | No of sequences | No of annotations | Backbone | Method | Epochs | Parameters | Training time (sec) | Backbone | Epochs | Parameters | Training time (sec) |
| Image captioning | | | | 46,880 | | | | lr: 0.0001 Op: Adam | | | | lr: 0.0001, Op: Adam, | 5000 |
| VQA | 16 | 10 | 4688 | 2,34,400 | SC-VAE | Latent data interpolation | 100 | Loss: KL divergence, Reconstruction B:16 | 1050 | MobileNetV2 +LSTM | 50 | Loss: Cross entropy, I: 224 × 224, B: 64 | 5430 |

This efficiency is critical for meeting the real-time constraints of autonomous vehicles. While other feature extractors like VGG16 and ResNet are powerful, their computational intensity and larger model sizes render them unsuitable for this application. The MobileNetV2's design is particularly effective at extracting salient, high-level features resilient to the visual degradation inherent in transitional weather. Our SCA module is built upon LSTM networks for text processing and sequential context modelling. LSTMs are uniquely suited for this task due to their ability to model long-range dependencies and maintain a robust contextual state over time, which is essential for both image captioning and VQA. They effectively overcome the vanishing gradient problem that limits the performance of standard RNNs on longer sequences, ensuring the model can maintain a coherent understanding of a full sentence or a series of frames. While GRUs offer a more lightweight alternative, their reduced flexibility makes them less suitable for the intricate context modeling required to generate precise and detailed textual descriptions of dynamic weather scenarios. The detailed computational and performance analyses of these architectural choices are provided in the next section.

## 5 Experiments

### 5.1 Dataset and Metrics

Table 3 summarizes the AIWD16-text dataset, outlining the transition states, annotation details, and implementation settings used for data generation and VLAAD-TW, including backbone, methodology, number of epochs, runtime, and hyperparameters. Transitional weather images for the AIWD16-text dataset were generated by a stochastic conditional VAE (SC-VAE), which was utilized to process pairs of input images with distinct weather conditions and perform latent data interpolation. Figure 5 shows a smooth latent representation of SC-VAE when presented with various extreme weather conditions, visually confirming its ability to learn a continuous and structured latent space. These input image pairs were collected from the Weather Driving (WD) dataset, including five weather conditions, sunny, cloudy, rainy, foggy, and snowy. Cloudy images were sourced from Cityscapes (Cordts et al., 2016), rainy images from RainCityscapes (Hu et al., 2019) & MultiWeatherCity (Muşat et al., 2021), and foggy images from Foggy Cityscapes (Sakaridis et al., 2018). Snowy images are generated using ControlNet (Zhang et al., 2023a), a stable diffusion-based model. After addressing the class imbalance, each weather class in the WD dataset contains 293 images, resized to a resolution of $512 \times 256$ pixels. The final AIWD16-text dataset comprises 4,688 weather transition sequences, each uniformly sampled at a length of $\mathcal{T} = 10$ frames, covering sixteen distinct weather transition states, namely, cloudy to rainy (CR), sunny to foggy (SF), sunny to rainy (SR), cloudy to snowy (CSn), cloudy to foggy (CF), snowy to rainy (SnR), snowy to foggy (SnF), foggy to rainy (FR) and vice versa. The dataset's total size is 7 GB. Figures 6, 7, 8, and 9 present the qualitative results of the generated transitional weather sequences. The proposed SC-VAE synthesizes smooth progressions between distinct weather conditions (e.g., cloudy, rainy, foggy, snowy, and sunny), demonstrating its ability to capture continuous and realistic visual transitions across a specified sequence length $T$.

**Dataset annotations.** Our dataset annotation process encompassed two primary tasks: image captioning and visual question-answering (VQA). We manually created textual descriptions for the image captioning component that captured both environmental and traffic elements. The captions specifically documented current weather conditions and their intensity levels while also describing comprehensive road scene information, including moving vehicles and pedestrians in front of the ego vehicle and vehicles parked along the roadside. Also, we developed a VQA dataset by creating five carefully crafted open-ended question-answer pairs for each image. These questions were designed to probe various aspects of the road scene, focusing on key information such as the current weather and its intensity, the number and types of moving and parked vehicles, and detailed pedestrian-related information.

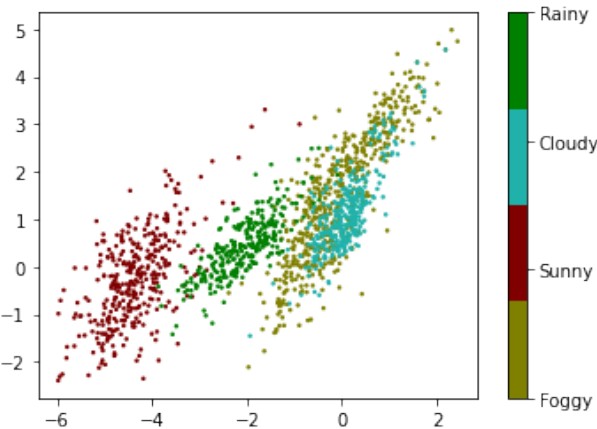

Figure 5: Latent space representation of the input images generated by C-VAE. The representation is smooth and continuous.

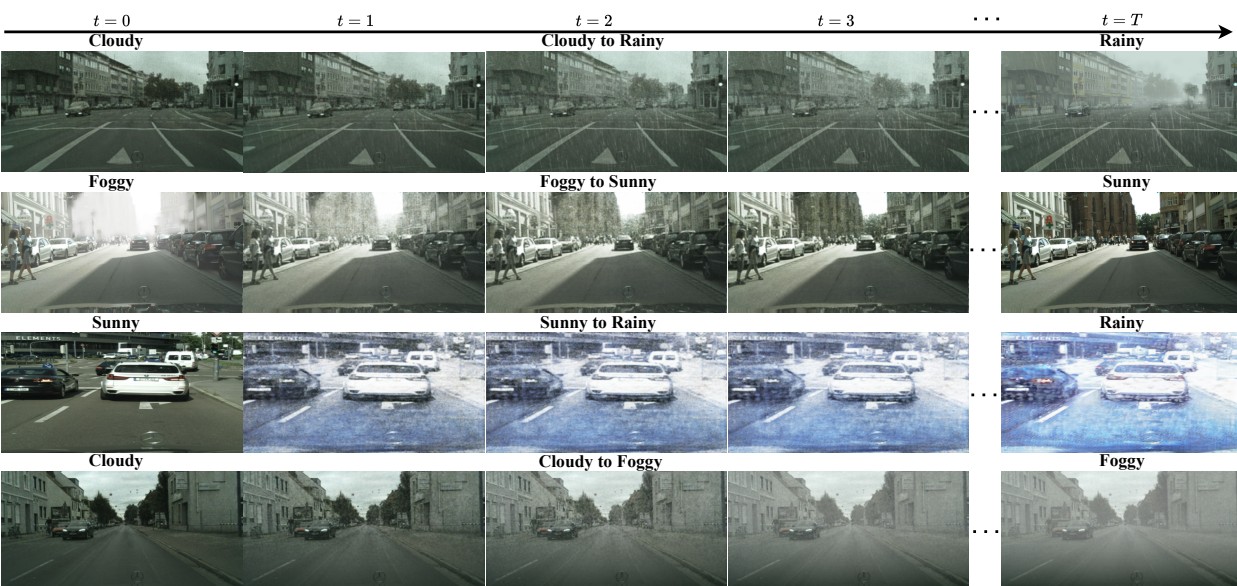

Figure 6: Qualitative results of SC-VAE generated transitional weather sequences (e.g., cloudy to rainy, foggy to sunny, sunny to rainy, and cloudy to foggy). For a given length $T$, the model produces smooth and realistic transitions between weather conditions.

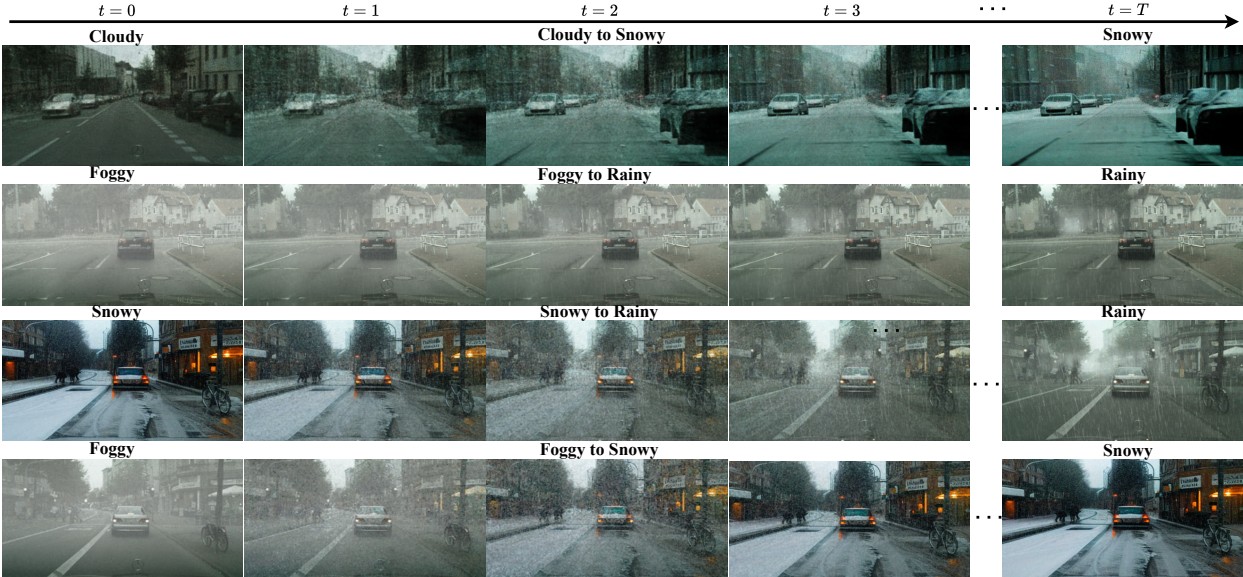

Figure 7: Qualitative results of SC-VAE generated transitional weather sequences (e.g., cloudy to snowy, foggy to rainy, snowy to rainy, and foggy to snowy). For a given length $T$, the model produces smooth and realistic transitions between weather conditions.

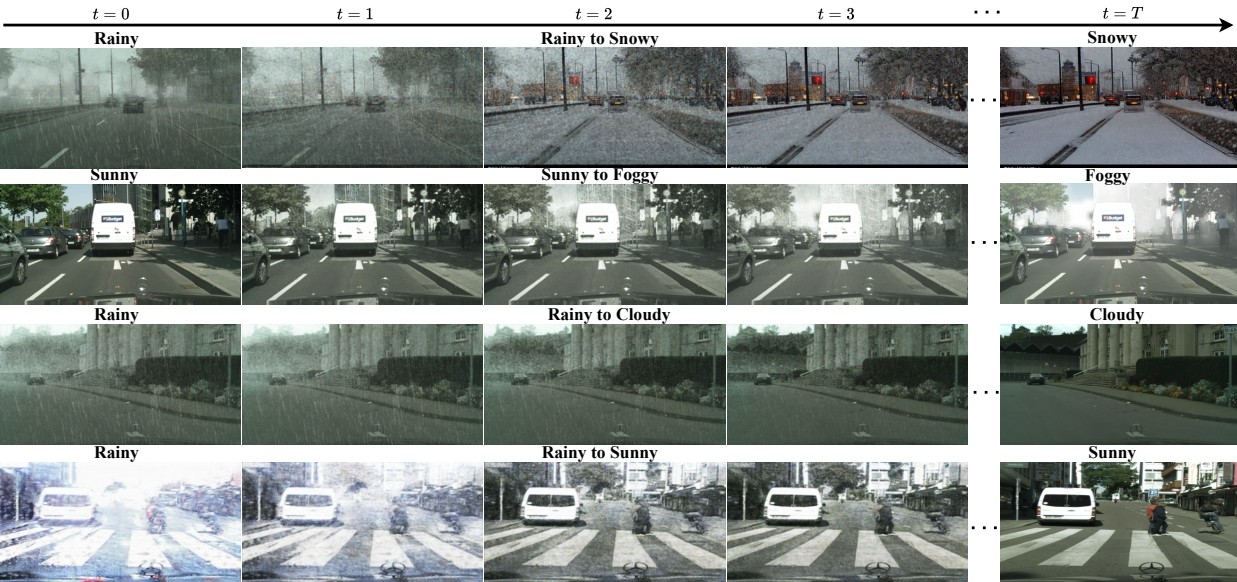

Figure 8: Qualitative results of SC-VAE generated transitional weather sequences (e.g., rainy to snowy, sunny to foggy, rainy to cloudy, and rainy to sunny). For a given length $T$, the model produces smooth and realistic transitions between weather conditions.

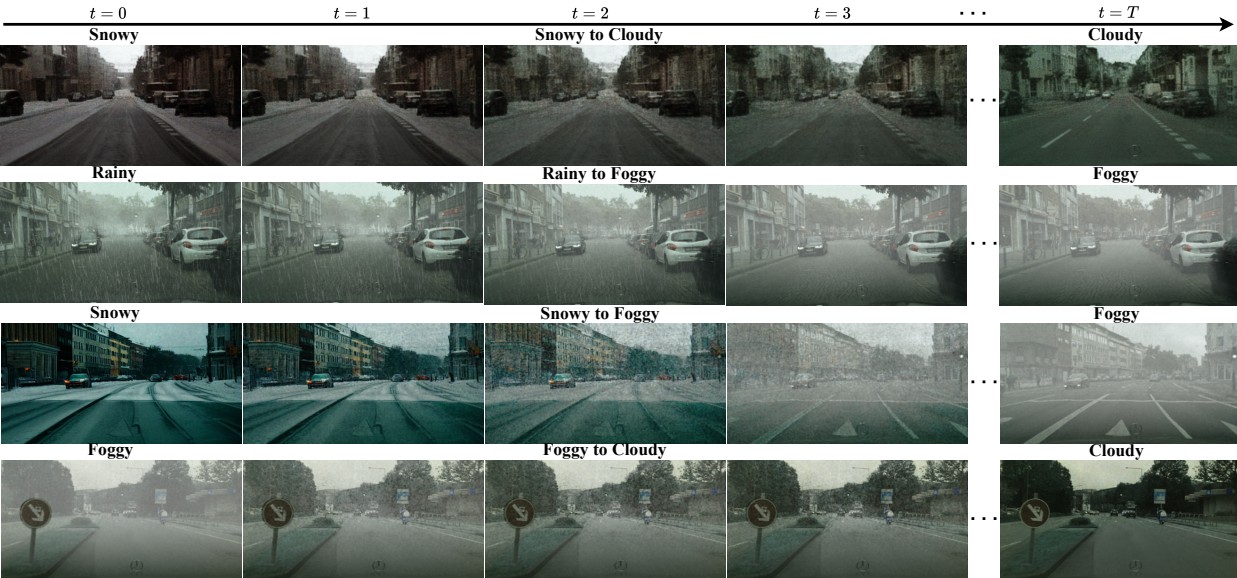

Figure 9: Qualitative results of SC-VAE generated transitional weather sequences (e.g., snowy to cloudy, rainy to foggy, snowy to foggy, and foggy to cloudy). For a given length $T$, the model produces smooth and realistic transitions between weather conditions.

**Evaluation metrics.** To evaluate the performance of image captioning and VQA tasks compared to other methods, we use the BLEU (Papineni et al., 2002) and ROUGE (Lin, 2004) scores. The BLEU score measures the precision of word sequences (n-grams) that match the generated text and ground truth. It is computed as

$$\text{BLEU} = BP \cdot \exp \left( \frac{1}{n} \sum_{i=1}^{n} \log p_i \right).$$

Where $BP$ is brevity penalty and $p_i$ is the precision of $i$-grams (in this work we use BLEU-4 i.e., i=4). ROUGE-L evaluates the semantic similarity between the generated and ground truth texts using the longest common subsequence, and it is computed as.

$$\text{ROUGE-L} = \frac{(1 + \gamma^2) \cdot \text{Recall} \cdot \text{Precision}}{\gamma^2 \cdot \text{Recall} + \text{Precision}}.$$

Recall is the fraction of the ground truth tokens in the predicted text. Precision is the fraction of the predicted text's tokens in the ground truth, and $\gamma$ is the weighting factor (default is 1) to balance the importance of recall and precision. Here, ROUGE-1 (L=1) measures the overlap of unigrams (single words) between the generated and ground truth texts, and ROUGE-2 (L=2) measures the overlap of bigrams (two consecutive words).

In addition, to evaluate the quality of the generated images, we use the Inception Score (IS). We also use Peak Signal-to-Noise Ratio (PSNR) and Signal-to-Noise Ratio (SNR) to quantify the noise levels within the dataset.

**Evaluating AIWD16-text dataset quality.** To ensure that the AIWD16-text dataset reflects real-world scenarios, we evaluate image quality using three key metrics: Inception Score (IS), Signal-to-Noise Ratio (SNR), and Peak Signal-to-Noise Ratio (PSNR). Table 4 presents a comparative analysis of these metrics across benchmark datasets. The results show that AIWD16-text consistently achieves higher IS, PSNR, and SNR values, highlighting its superior image quality and reliability as a resource for vision-language tasks, particularly under adverse weather conditions. While the generated transition images are generally effective, certain transitions such as snowy→rainy, rainy→snowy, sunny→rainy, and rainy→sunny, exhibit room for improvement. To further assess perceptual quality, we conducted a small-scale user study (N = 50), in which

Table 4: Quality of the AIWD16-text images measured using IS, PSNR, and SNR metrics. '-'→Not applicable. **Bold** indicates the best values.

| Dataset/ Metrics | BDD100K (Yu et al.) | Cityscapes (Cordts et al., 2016) | MW City (Musat et al., 2021) | CR | RC | SF | FS | SR | RS | CF | FC | CSn | SnC | SnR | RSn | SnF | FSn | FR | RF |
|---|---|---|---|---|---|---|---|---|---|---|---|---|---|---|---|---|---|---|---|
| IS ↑ | 4.51 | 5.00 | 4.13 | 54.70 | **58.66** | 49.39 | 54.91 | 58.28 | 48.17 | 50.44 | 51.11 | 49.95 | 48.55 | 46.55 | 45.34 | 48.34 | 47.55 | 52.47 | 51.92 |
| PSNR ↑ | - | 27.73 | 31.59 | 32.59 | 30.10 | 31.07 | **32.62** | 31.22 | 31.54 | 31.42 | 31.15 | 30.52 | 30.33 | 30.11 | 30.32 | 29.76 | 30.01 | 31.77 | 31.54 |
| SNR ↑ | - | - | - | 32.75 | 34.20 | 33.57 | **34.25** | 34.23 | 33.37 | 31.87 | 32.66 | 31.22 | 31.36 | 31.19 | 31.38 | 30.15 | 30.93 | 31.22 | 31.01 |
| Realism (1–5) ↑ | - | - | - | 4.55 | 4.5 | 4.23 | 4.25 | 3.60 | 3.52 | 4.16 | 4.02 | 3.95 | 3.87 | 3.33 | 3.24 | 3.75 | 3.67 | 3.99 | 3.91 |

Table 5: Comparison of the AIWD16-text dataset with other AVT datasets. ✓→Present, ✗→Not present, '-'→Not applicable.

| Dataset | No. of images | No. of captions | Image captioning | VQA | Adverse weather | Transitional weather |
|---|---|---|---|---|---|---|
| BDD-X (Kim et al., 2018) | 6,984 | 26,539 | ✓ | ✓ | ✓ | ✗ |
| HDD (Ramanishka et al., 2018) | 7,744 | - | ✗ | ✗ | ✓ | ✗ |
| CAP-DATA (Fang et al., 2022) | 11,727 | - | ✗ | ✗ | ✓ | ✗ |
| T2C (Deruyttere et al., 2019) | 850 | 11,959 | ✓ | ✗ | ✗ | ✗ |
| DRAMA (Malla et al., 2023) | 17,758 | 17,066 | ✗ | ✓ | ✓ | ✗ |
| NuScenes-QA (Qian et al., 2024) | 1,000 | 34,000 | ✗ | ✓ | ✗ | ✗ |
| DriveLM (Sima et al., 2025) | 4,072 | 3,77,983 | ✗ | ✗ | ✗ | ✗ |
| VLAAD (Park et al., 2024) | 10,379 | 64,000 | ✓ | ✓ | ✗ | ✗ |
| **AIWD16-text (Ours)** | **46,880** | **2,34,400** | ✓ | ✓ | ✓ | ✓ |

participants rated the realism of generated transitions on a 5-point Likert scale. The results (mean score = 3.90) indicate that most transitions were perceived as visually realistic, though a few (including rain → snow) received slightly lower ratings.

**Software and machine setup.** We used Python 3.8, PyTorch, and TensorFlow for our experiments on an NVIDIA Tesla M60 8GB GPU. Our VLAAD-TW model, trained for 50 epochs with a batch size of 64 and a learning rate of 0.0001, minimizes the cross-entropy loss function using the Adam optimizer. Also, our SC-VAE generation model trained for 100 epochs with a batch size of 16 and a learning rate of 0.0001 minimizes the KL divergence loss function using the Adam optimizer. We randomly split the dataset into 70% training, 10% validation, and 20% testing sets.

## 5.2 Comparison with Existing Vision Language Models.

Table 5 compares the proposed AIWD16-text dataset with other state-of-the-art VLM benchmark datasets for AVT. As observed, only a few datasets (Kim et al., 2018; Malla et al., 2023; Ramanishka et al., 2018; Fang et al., 2022) include images featuring adverse weather conditions relevant to AVT. Notably, DRAMA (Malla et al., 2023) and BDD-X (Kim et al., 2018) provide annotations related explicitly to adverse weather. However, the AIWD16 text dataset stands out as the first to feature transitional weather conditions, with images annotated for both VQA and captioning tasks, addressing a significant gap in existing benchmarks.

Tables 6 and 7 demonstrate the performance comparison between VLAAD-TW and state-of-the-art VLMs on the AIWD16-text dataset for image captioning and VQA tasks. We have selected these benchmark models for their diverse architectural approaches and proven capability in handling transitional weather conditions. Our analysis shows that VLAAD-TW achieves superior performance with a 2.01% improvement in BLEU score & 0.62% ROUGE-L metrics for image captioning and a 0.6% improvement in BLEU score & 0.83 in ROUGE-L for the VQA task, despite using significantly fewer parameters. The performance improvement stems from VLAAD-TW's lightweight architecture, which efficiently captures patterns in transitional weather conditions. In contrast, other foundation models exhibit lower performance due to overfitting on our dataset, highlighting their limited generalization in this context.

Table 6: Performance comparison between the proposed and state-of-the-art models for the image captioning task on the AIWD16-text Dataset. **Bold** indicates the best values.

| Model | BLEU-4 | ROUGE-1 | ROUGE-2 | ROUGE-L |
|---|---|---|---|---|
| BLIP (Li et al., 2022) | 39.52 | 38.63 | 27.34 | 37.55 |
| BLIP-2 (Li et al., 2023) | 43.69 | 44.95 | 31.93 | 43.10 |
| $EM-VLM4AD_{base}$ (Gopalkrishnan et al., 2024) | 38.54 | 39.23 | 26.66 | 38.88 |
| $EM-VLM4AD_{Q-large}$ (Gopalkrishnan et al., 2024) | 43.59 | 45.61 | 28.79 | 43.15 |
| DriveLM-Agent (Sima et al., 2025) | 45.12 | 47.36 | 29.55 | 42.31 |
| **VLAAD-TW (Ours)** | **47.13** | **49.29** | **31.99** | **43.77** |

Table 7: Performance comparison between the proposed and state-of-the-art models for the VQA task on the AIWD16-text Dataset. **Bold** indicates the best values.

| Model | Acc. | BLEU-4 | ROUGE-1 | ROUGE-2 | ROUGE-L |
|---|---|---|---|---|---|
| BLIP (Li et al., 2022) | 49.76 | 34.44 | 36.05 | 24.34 | 30.55 |
| BLIP-2 (Li et al., 2023) | 55.21 | 41.23 | 42.68 | 29.18 | 38.40 |
| $EM-VLM4AD_{base}$ (Gopalkrishnan et al., 2024) | 49.21 | 39.27 | 45.61 | 28.79 | 39.15 |
| $EM-VLM4AD_{Q-large}$ (Gopalkrishnan et al., 2024) | 45.77 | 37.95 | 44.23 | 26.66 | 38.88 |
| DriveLM-Agent (Sima et al., 2025) | 52.45 | 39.99 | 43.67 | 27.45 | 38.69 |
| **VLAAD-TW (Ours)** | **55.90** | **41.83** | **45.80** | **29.87** | **39.98** |

Table 8 presents the performance of VLAAD-TW, pretrained on the AIWD16-text dataset and fine-tuned on Drive-LM. The results demonstrate the effectiveness of VLAAD-TW, achieving competitive BLEU and ROUGE scores on the Drive-LM benchmark.

Despite significantly fewer parameters. This fine-tuning performance of VLAAD-TW on the DriveLM dataset indicates that pretraining on the proposed AIWD16-text dataset effectively enhances the model's visual–language understanding and transferability. This demonstrates that the learned representations generalize well beyond the pretraining domain, improving performance on downstream driving-related tasks.

Table 9 presents the performance of VLAAD-TW, which is pre-trained on the AIWD16-text dataset and subsequently fine-tuned on the nuScenes dataset to evaluate its transferability and robustness on VQA task. The results show that VLAAD-TW (AIWD16-text→nuScenes) achieves notable improvements over existing baselines, including OmniDrive, across multiple metrics such as accuracy, BLEU, ROUGE-L and CIDEr scores. These findings demonstrate that VLAAD-TW pretraining on AIWD16-text effectively enhances the model's ability to adapt to diverse and evolving driving scenarios, thereby validating both the effectiveness of VLAAD-TW and the practical utility of the AIWD16-text dataset. **Computational analysis.** We evaluated the memory and computational efficiency of VLAAD-TW, crucial factors for real-time systems with resource constraints. Table 10 compares computational complexity metrics between VLAAD-TW and

Table 8: Performance comparison of proposed and state-of-the-art models on the DriveLM VQA task. **Bold** indicates the best values.

| Model | BLEU-4 | ROUGE-1 | ROUGE-2 | ROUGE-L |
|---|---|---|---|---|
| BLIP (Li et al., 2022) | 34.44 | 36.05 | 24.34 | 30.55 |
| BLIP-2 (Li et al., 2023) | 41.23 | 42.68 | 29.87 | 38.98 |
| $EM-VLM4AD_{base}$ (Gopalkrishnan et al., 2024) | 45.36 | 57.77 | 29.11 | 71.98 |
| $EM-VLM4AD_{Q-large}$ (Gopalkrishnan et al., 2024) | 40.11 | 55.91 | 27.85 | 70.72 |
| DriveLM-Agent (Sima et al., 2025) | 53.09 | 56.11 | 28.99 | 66.79 |
| **VLAAD-TW (Ours)** | **53.81** | **58.04** | **29.15** | **72.45** |

Table 9: Performance comparison of proposed and state-of-the-art models pretrained on the AIWD16-text dataset and finetuned on the NuScenes Dataset. **Bold** indicates the best values. '-'→Not applicable.

| Model | Acc. | BLEU-4 | ROUGE-L | CIDEr |
|---|---|---|---|---|
| Omni-L (Wang et al., 2025) | 53.27 | 42.45 | 61.55 | 73.89 |
| Omni-Q (Wang et al., 2025) | 52.15 | - | - | 68.86 |
| OPT (Zhang et al., 2022) | - | 38.80 | 47.70 | - |
| $EM-VLM4AD_{base}$ (Gopalkrishnan et al., 2024) | - | 42.36 | 68.99 | - |
| LLAMA-Adapter (Zhang et al., 2024b) | - | 43.17 | 67.56 | - |
| **VLAAD-TW (Ours)** | **57.96** | **45.73** | **69.09** | **75.62** |

Table 10: Computational comparison between the proposed model and state-of-the-art VLMs. '-'→Not applicable, **Bold** indicates the best values.

| Model | Vision Backbone | Text Backbone | No. of Param | FLOPs | Memory (GB) | FPS | Time (ms) |
|---|---|---|---|---|---|---|---|
| $EM-VLM4AD_{base}$ (Gopalkrishnan et al., 2024) | ViT | T5-Base | 235M | 9.47B | 0.94 | 2.9 | 635 |
| $EM-VLM4AD_{Q-large}$ (Gopalkrishnan et al., 2024) | ViT | T5-Large | 769M | 31.5B | 0.77 | 1.2 | 833 |
| DriveLM-Agent (Sima et al., 2025) | LoRA, BLIP-2 | | 3.96B | 439B | 14.43 | 0.16 | 6250 |
| LLM-Driver (Chen et al., 2024a) | LLaMA-7b | | 7B | 268B | 28 | - | - |
| BLIP (Li et al., 2022) | ViT | BERT | 300M | 4.2B | 0.95 | 4.9 | 204 |
| BLIP-2 (Li et al., 2023) | CLIP ViT | Frozen LLM | 4.1B | 23B | 15 | 2.1 | 476 |
| **VLAAD-TW (Ours)** | **CNN** | **LSTM** | **12M** | **810M** | **0.25** | **11.2** | **89.3** |

state-of-the-art VLMs for AVT across five key indicators: model parameters (param), FLOPs, memory utilization, FPS and inference speed (time). VLAAD-TW demonstrates remarkable efficiency, requiring only 12M parameters, 810M FLOPs, and 0.25GB of memory. Compared to the current foundation VLMs, this reflects an 85% reduction in parameters, 80% reduction in FLOPs, and 73% reduction in memory usage. These significant improvements streamline the architecture, enable faster inference and higher frames per second (FPS), and greatly reduce resource consumption while maintaining competitive performance. Our model achieves substantially higher FPS and inference speed compared to other models. Such optimizations make VLAAD-TW particularly well-suited for resource-constrained environments and real-time applications where rapid inference and minimal resource consumption are essential.

**Qualitative analysis.** Figures 10 and 11 show qualitative results for image captioning and VQA across sunny to foggy, cloudy to rainy, sunny to rainy weather transitions, highlighting VLAAD-TW's ability to interpret scenes and generate accurate captions and answers under challenging conditions. Despite strong overall performance, certain limitations were observed. The model encounters difficulties perceiving small and distant road elements during moderate to heavy rain and fog, primarily due to varying and low illumination conditions. Additionally, the model occasionally misinterprets subtle weather variations. These challenges will be addressed in future work.

# 6 Ablation Study

**Model performance across various transition states.** Table 11 summarizes the image captioning performance of the proposed VLAAD-TW model across various transitions in the AIWD16-text dataset. The Rainy to Cloudy, and Cloudy to Rainy transitions yield the best results. Similarly, Table 12 presents the VQA performance, where these transitions outperform others. We performed an additional ablation study to assess the impact of transitional weather modeling by comparing VLAAD-TW trained on (a) WD (Weather Driving) alone and (b) AIWD16-text. The results in Table 13 show that training with AIWD16-text leads to substantial improvements across all evaluation metrics, indicating that transitional weather data provides more informative supervision.

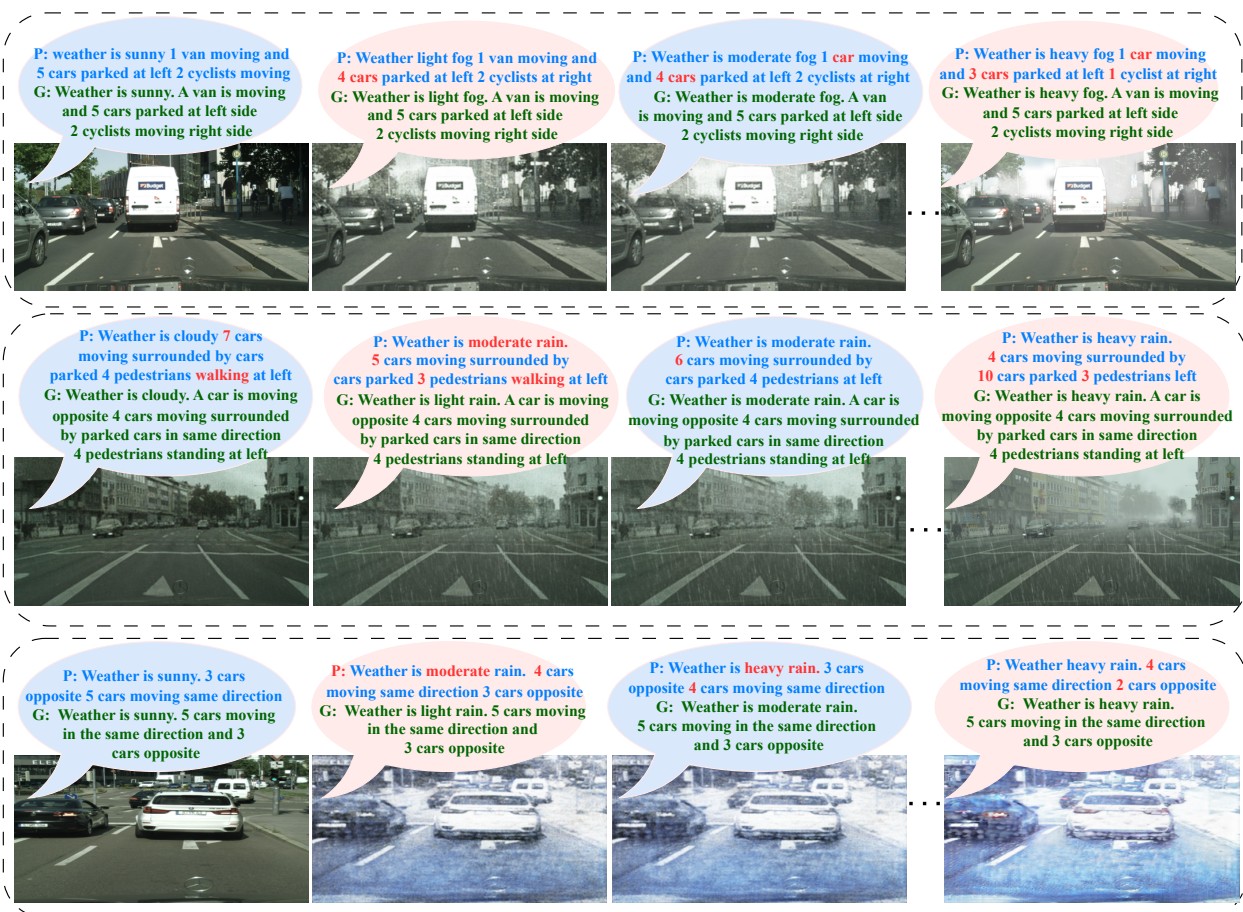

Figure 10: Qualitative analysis of the proposed VLAAD-TW approach for image captioning on the AIWD16-text dataset. Green highlights the ground truth, Blue highlights the predicted text, and Red highlights the wrong prediction. Q→Query, P→Prediction, G→Ground truth.

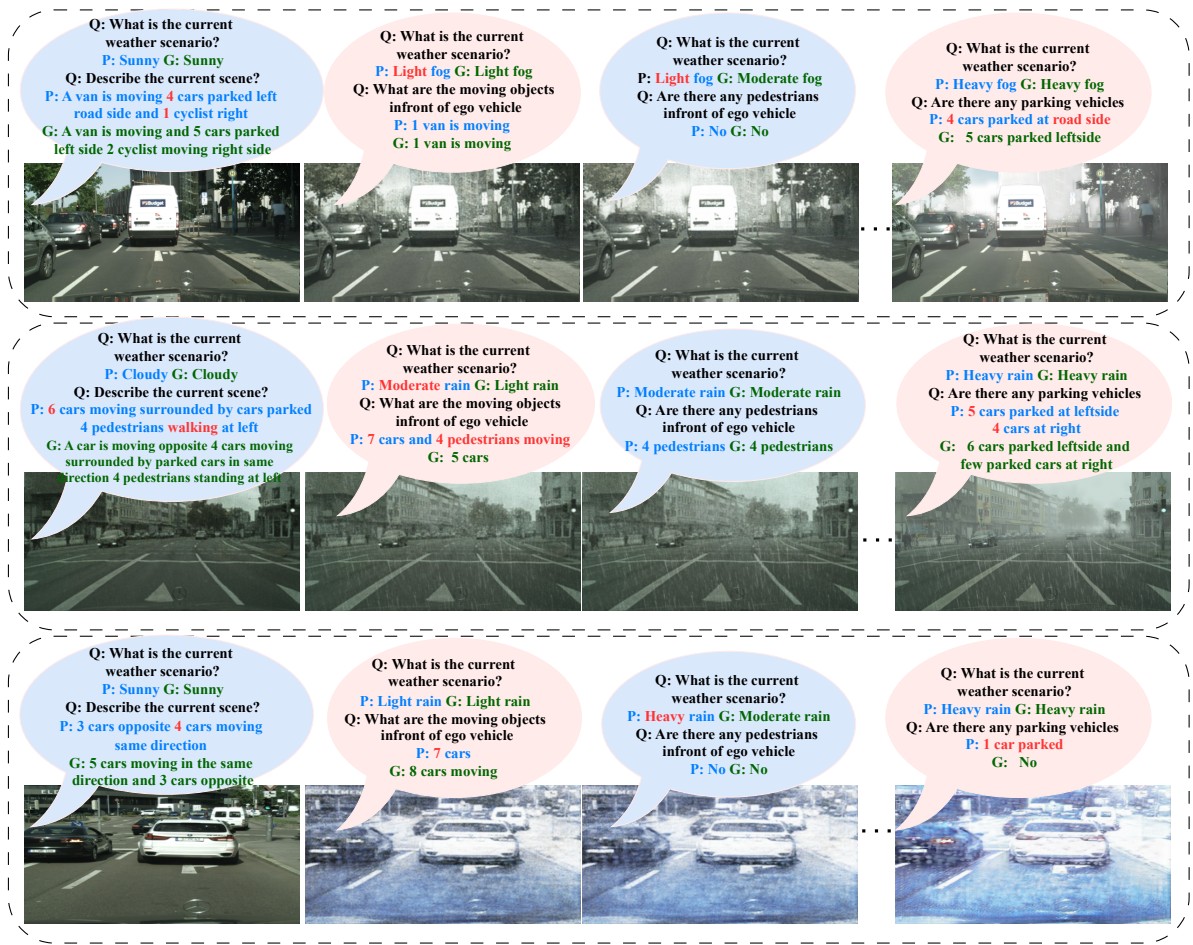

Figure 11: Qualitative analysis of the proposed VLAAD-TW approach for VQA on the AIWD16-text dataset. Green highlights the ground truth, Blue highlights the predicted text, and Red highlights the wrong prediction. Q→Query, P→Prediction, G→Ground truth.

Table 11: Performance of the proposed model for image captioning on each transition state of the AIWD16-text dataset. **Bold** indicates the best values.

| Transition/ Metric | RC | CR | SF | FS | SR | RS | CF | FC | CSn | SnC | SnR | RSn | SnF | FSn | FR | RF |
|---|---|---|---|---|---|---|---|---|---|---|---|---|---|---|---|---|
| BLEU-4 | 49.24 | **50.07** | 48.19 | 47.91 | 38.27 | 38.16 | 48.12 | 47.41 | 43.26 | 42.01 | 38.33 | 37.89 | 40.65 | 40.92 | 48.22 | 47.40 |
| ROUGE-1 | 51.33 | **54.65** | 50.10 | 49.05 | 41.35 | 40.98 | 50.45 | 48.91 | 46.11 | 45.99 | 40.76 | 40.23 | 41.98 | 42.85 | 50.16 | 49.54 |
| ROUGE-2 | 38.18 | **39.23** | 34.44 | 32.61 | 28.67 | 27.55 | 36.26 | 35.98 | 33.05 | 33.16 | 27.44 | 27.01 | 29.78 | 31.46 | 37.33 | 37.16 |
| ROUGE-L | 47.16 | **48.07** | 39.78 | 37.25 | 30.12 | 29.87 | 47.12 | 46.55 | 42.62 | 42 | 29.79 | 29.72 | 35.90 | 37.15 | 44.71 | 44.11 |

Table 12: Performance of the proposed model for VQA on each transition state of the AIWD16-text dataset. **Bold** indicates the best values.

| Transition/ Metric | RC | CR | SF | FS | SR | RS | CF | FC | CSn | SnC | SnR | RSn | SnF | FSn | FR | RF |
|---|---|---|---|---|---|---|---|---|---|---|---|---|---|---|---|---|
| Accuracy | 58.67 | **59.65** | 49.20 | 48.76 | 42.17 | 41.05 | 56.76 | 55.25 | 44.77 | 43.28 | 42.11 | 43.23 | 45.60 | 45.77 | 48.33 | 47.37 |
| BLEU-4 | 45.80 | **47.92** | 44.45 | 42.24 | 35.79 | 35.23 | 44.15 | 43.25 | 39.19 | 38.42 | 36.43 | 37.03 | 39.77 | 39.99 | 43.15 | 42.94 |
| ROUGE-1 | 49.21 | **50.35** | 46.36 | 45.11 | 38.89 | 38.87 | 47.49 | 45.96 | 40.21 | 40.18 | 39.89 | 40.12 | 41.76 | 40.88 | 46.06 | 45.91 |
| ROUGE-2 | 32.66 | **34.88** | 30.55 | 31.90 | 25.97 | 26.68 | 31.45 | 31.17 | 29.05 | 28.50 | 26.12 | 26.96 | 29.24 | 29.47 | 30.55 | 31.10 |
| ROUGE-L | 44.76 | **46.54** | 35.78 | 34.48 | 28.65 | 29.11 | 40.33 | 38.65 | 39.62 | 38.88 | 28.87 | 29.66 | 34.36 | 34.64 | 36.28 | 35.76 |

**Computational and performance analysis across various feature extractor backbones.** Table 14 details the computational complexity of the proposed VLAAD-TW using VGG16 (Simonyan & Zisserman, 2014) and ResNet50 (He et al., 2016) as feature extractors. Compared to MobileNetV2, VGG16 requires $7.2\times$ more memory, $19\times$ more FLOPs, and $11.5\times$ more parameters, while ResNet50 demands $4.2\times$ more memory, $5\times$ more FLOPs, and $3.6\times$ more parameters. The table also highlights the performance of these feature extractors, showing that VLAAD-TW with MobileNetV2 outperforms others in terms of BLEU and ROUGE scores for the image captioning task. Figure 12 provides a qualitative comparison of the proposed VLAAD-TW model across different feature extractors for the VQA. For each image–question pair, answers produced by VLAAD-TW using various visual backbones are shown to illustrate how the choice of feature extractor influences semantic grounding, object recognition, and reasoning accuracy.

**Performance analysis across various sequence model backbones.** Table 15 reports the performance of the proposed VLAAD-TW using three recurrent sequence models, GRU (Chung et al., 2014), RNN (Elman, 1990), and LSTM (Hochreiter & Schmidhuber, 1997). As shown in the table, the LSTM variant achieves the best overall performance among the evaluated models. This observation is further supported by the qualitative examples in Figure 13, where the LSTM-based model provides more accurate and contextually appropriate answers. These results indicate that increasingly expressive recurrent units enhance question understanding and multimodal reasoning in VQA. Overall, the analysis supports the design choice of pairing the VLAAD-TW architecture with efficient components such as the MobileNetV2 feature extractor and the LSTM sequence model.

**Ablation of SAFM and SCA Modules.** Furthermore, we conduct two module-level ablations to isolate the contribution of the higher-level components of our architecture. First, in the w/o SAFM variant, we remove the attention-guided fusion block and instead form global visual and textual descriptors using only the final hidden states of the image and question encoders. These vectors are concatenated and passed through a fully connected layer, thereby eliminating all cross-modal attention and fine-grained relevance weighting. Second, in the w/o SCA configuration, we disable the entire Spatiotemporal Context Aggregator and rely solely on a single-pass text encoder to obtain the question representation, removing the additional temporal reasoning and contextual enhancement normally provided by SCA. These ablations allow us to

Table 13: Performance comparison demonstrating the benefit of training VLAAD-TW with AIWD16-text rather than WD alone. **Bold** indicates the best values.

| Dataset | BLEU | ROUGE-1 | ROUGE-2 | ROUGE-L |
|---|---|---|---|---|
| WD | 37.50 | 37.40 | 20.28 | 33.33 |
| AIWD16-text | 47.13 | 49.29 | 31.99 | 43.77 |

Table 14: Computational and performance comparison of the proposed VLAAD-TW with different feature extractors. **Bold** indicates the best values.

| Feature extractor | No. of parameters | FLOPs | Memory | BLEU-4 | ROUGE-1 | ROUGE-2 | ROUGE-L |
|---|---|---|---|---|---|---|---|
| VGG16 (Simonyan & Zisserman, 2014) | 138M | 15.5B | 1.8 | 41.24 | 40.09 | 23.48 | 37.21 |
| Resnet50 (He et al., 2016) | 50M | 4.2B | 0.9 | 44.47 | 39.29 | 24.55 | 36.42 |
| MobileNetV2 (Sandler et al., 2018) | **12M** | **810M** | **0.25** | **41.83** | **45.80** | **29.87** | **39.98** |

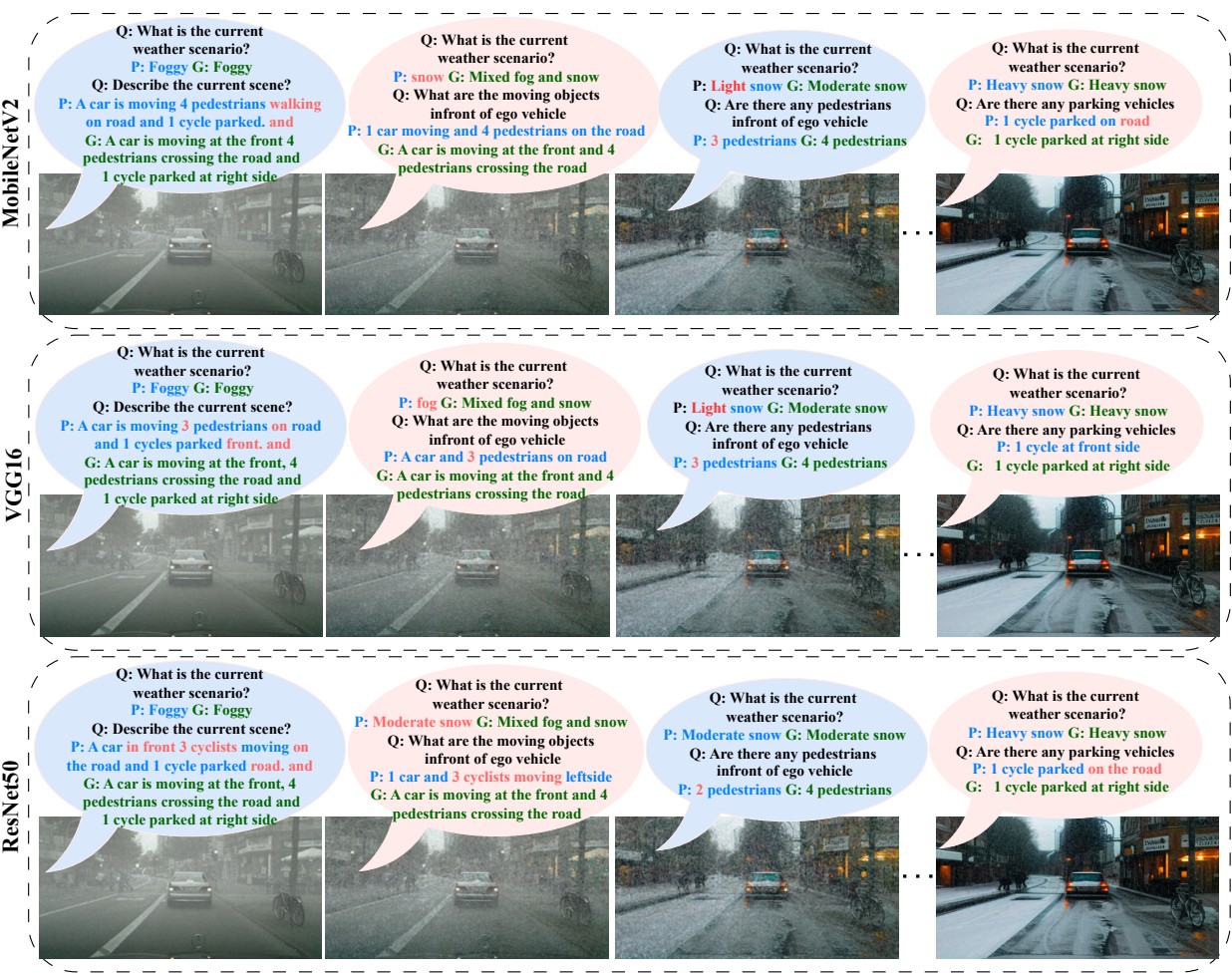

Figure 12: Qualitative evaluation showing how different feature extractors affect the performance of the proposed VLAAD-TW model. Green highlights the ground truth, Blue highlights the predicted text, and Red highlights the wrong prediction. Q→Query, P→Prediction, G→Ground truth.

Table 15: Performance comparison of the proposed VLAAD-TW with different sequence models. **Bold** indicates the best values.

| Sequence model | BLEU-4 | ROUGE-1 | ROUGE-2 | ROUGE-L |
|---|---|---|---|---|
| GRU (Chung et al., 2014) | 38.39 | 39.21 | 22.23 | 30.61 |
| RNN (Elman, 1990) | 28.65 | 36.21 | 18.99 | 26.78 |
| LSTM (Hochreiter & Schmidhuber, 1997) | **41.83** | **45.80** | **29.87** | **39.98** |

Table 16: Ablation analysis showing the effect of removing the SAFM and SCA modules on AIWD16-text for the VQA task.

| Variant | BLEU-4 | ROUGE-1 | ROUGE-2 | ROUGE-L |
|---|---|---|---|---|
| W/o SAFM | 34.92 | 36.29 | 24.65 | 33.72 |
| W/o SCA | 38.25 | 40.96 | 27.59 | 36.66 |
| VLAAD-TW (Full model) | **41.83** | **45.80** | **29.87** | **39.98** |

directly quantify the contribution of SAFM and SCA to robust scene understanding and descriptive quality, as summarized in Table 16 and illustrated qualitatively in Figure 14.

**Generalization of VLAAD-TW on unseen weather domains.** Table 17 reports the performance of VLAAD-TW on unseen weather domains. The model was trained on the Cloudy to Rainy transition and evaluated on other transition scenarios. A noticeable drop in performance highlights the domain gap between training and testing conditions. Figure 15 further illustrates this effect by showing qualitative results when the model is trained on the AIWD16-text dataset and tested on the unseen BDD100K dataset. As ground-truth VQA annotations are unavailable for BDD100K, the incorrect predictions were identified through visual inspection. We plan to address this domain discrepancy in future work using unsupervised domain adaptation techniques.

## 6.1 Discussion on Edge Cases

**Unusual weather patterns.** The training data incorporates transitional weather scenarios of varying intensity to strengthen model robustness across conditions. Using the SC-VAE, we perform controlled data interpolation with the parameter $\beta \in [0,1]$ to generate a continuous spectrum of weather transitions, from $t = 0$ to $T$. This methodology allows us to synthesize an extensive dataset that captures various weather variations. This diverse training data enables our model to adapt effectively while sustaining high performance across a wide range of scenarios. However, ensuring optimal performance in highly unusual or unforeseen weather patterns remains a challenge, presenting a direction for future work to further strengthen VLAAD-TW's adaptability and ensure reliable performance across a wider range of environmental conditions.

**Performance on real-world noise and adversarial robustness.** Our model performs well across various challenging weather conditions, and real-world noise and adversarial perturbations may introduce additional challenges. Weather transitions often exhibit irregular patterns, where certain dependencies are more critical than others. To further strengthen robustness, we plan to explore adversarial defense strategies, including

Table 17: Quantitative results of the proposed VQA model trained on the Cloudy to Rainy transition and tested on other weather transitions to evaluate generalization to unseen weather domains. **Bold** indicates the best values.

| Transition/ Metric | RC | SF | FS | SR | RS | CF | FC | CSn | SnC | SnR | RSn | SnF | FSn | FR | RF |
|---|---|---|---|---|---|---|---|---|---|---|---|---|---|---|---|
| Accuracy | **49.12** | 42.26 | 42.44 | 36.99 | 35.90 | 41.56 | 41.22 | 40.56 | 39.88 | 35.81 | 35.22 | 37.42 | 36.21 | 40.37 | 40.07 |
| BLEU-4 | 39.66 | 38.59 | 37.29 | 31.65 | 29.97 | **39.67** | 39.28 | 35.28 | 34.96 | 30.34 | 30.89 | 33.17 | 33.34 | 36.85 | 36.17 |
| ROUGE-1 | **44.77** | 38.60 | 37.25 | 33.98 | 31.22 | 40.10 | 39.27 | 36.12 | 35.78 | 31.10 | 31.77 | 34.66 | 34.56 | 37.10 | 36.91 |
| ROUGE-2 | **31.07** | 28.14 | 28.11 | 24.79 | 24.97 | 28.44 | 28.88 | 25.05 | 24.95 | 23.15 | 23.12 | 25.93 | 25.45 | 28.43 | 27.89 |
| ROUGE-L | **40.22** | 35.18 | 31.83 | 26.08 | 26.65 | 36.85 | 33.33 | 31.66 | 30.52 | 25.27 | 26.66 | 32.97 | 32.40 | 33.37 | 32.66 |

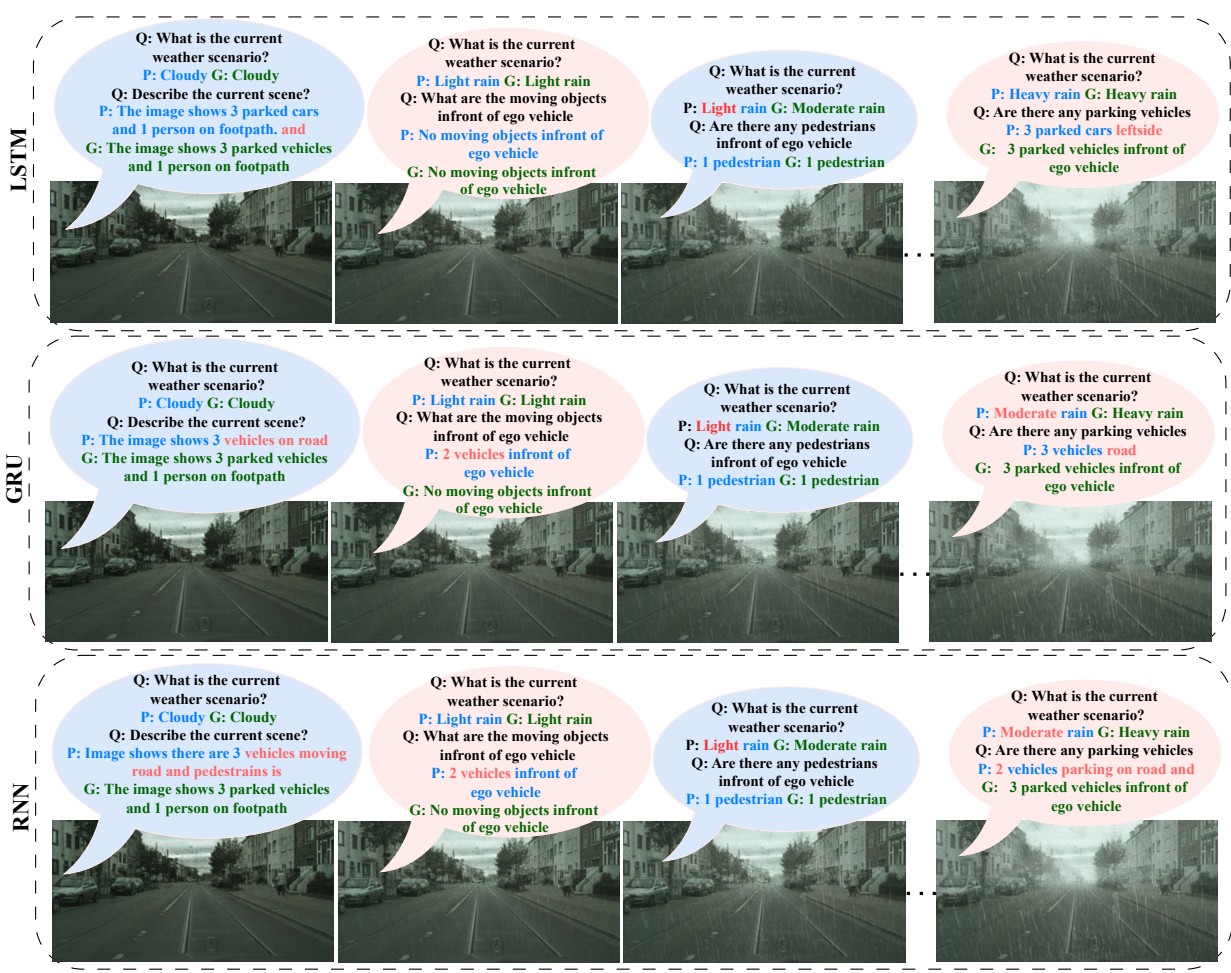

Figure 13: Qualitative evaluation of the proposed VLAAD-TW method using different sequence models on VQA examples. Green highlights the ground truth, Blue highlights the predicted text, and Red highlights the wrong prediction. Q→Query, P→Prediction, G→Ground truth.

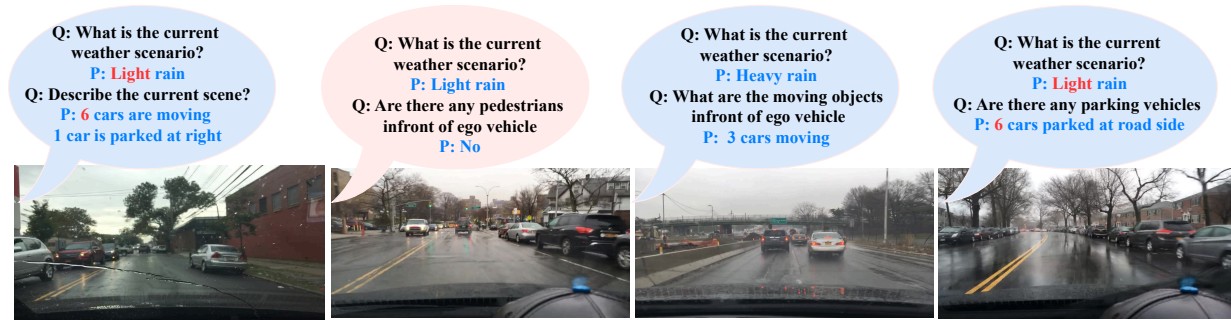

Figure 14: Qualitative analysis showing the effect of removing the SAFM and SCA modules on AIWD16-text. Green highlights the ground truth, Blue highlights the predicted text, and Red highlights the wrong prediction. Q→Query, P→Prediction, G→Ground truth.

Figure 15: Qualitative analysis of the proposed VLAAD-TW model trained on the AIWD16-text dataset and tested on the unseen BDD100K dataset. The results illustrate prediction errors caused by domain discrepancies with visual inspection, as ground-truth VQA annotations are unavailable.

adversarial training and uncertainty quantification techniques, to mitigate potential vulnerabilities. These enhancements will ensure our model maintains strong generalization across real-world weather conditions.

## 6.2 Potential Biases of AIWD16-text Dataset

While the AIWD16-text dataset provides diverse intensities of weather conditions, it has certain limitations and potential biases, which we intend to investigate further in future work. *(i)* Limited geographic representation: The dataset contains static background scenes throughout a weather transition sequence, representing only a limited geographic area. This design enables a controlled and interpretable setting for evaluating vision-language models and perception systems across weather transitions. By isolating weather variations from background motion, our method provides a reliable mechanism for analyzing model robustness to transitional weather. In future work, we plan to explore video-based weather transition models to capture dynamic scene background along with weather transitions. *(ii)* Temporal sampling bias: The dataset captures transitions at 10 frames per second, which may not adequately represent all weather intensity variations. This fixed sampling rate could introduce bias by missing intermediate weather states. In future work, we will use different sampling rates and create annotations for them. *(iii)* Potential biases in data division: The dataset is divided randomly into training, validation, and testing sets. However, this approach may not account for biases in the data distribution, such as variations in the number of images of various weather scenarios, traffic scenarios or road elements. *(iv)* Annotation biases: The accuracy and consistency of the annotations for visual interpretation could be subject to human biases. This could lead to systematic under-representation or misclassification of certain road elements, which may impact the model's ability to generalize to real-world scenarios. Evaluating the quality and potential biases in the road element textual annotations should be a focus of future investigations.

## 6.3 Unique Advantages of SC-VAE Weather Transition Generation

While latent space interpolation is a known generative technique, our primary contribution lies in its novel and task-specific application to synthesize a continuous and diverse range of transitional weather states. Existing generative methods often focus on discrete attribute manipulation or simple style transfer. In contrast, our approach models the seamless and progressive evolution between complex weather conditions, directly addressing the critical lack of continuous weather data for robust VLM training in AVT. This adaptation is not a trivial reuse of existing methods but a purpose-built, domain-aware methodology designed to solve a significant, real-world challenge in autonomous driving perception. To the best of our knowledge, this is the first work to successfully apply data interpolation to generate transitional weather data, making it a distinct and timely contribution to the field.

## 7 Limitations and Future Work

Despite the effectiveness of the VLAAD-TW, the current work has certain limitations that we plan to address in future research.

1. **Background consistency and other possible weather transitions.** While the generated images depict realistic scenarios, the backgrounds remain static across weather changes due to the use of input image pairs with consistent backgrounds. This design enables accurate evaluation of vision-language tasks under varying weather conditions. Future work will explore transitional video generation with dynamic backgrounds while preserving fine-grained weather attributes.

2. **Long-range dependency.** While VLAAD-TW excels in real-time AVT tasks and outperforms other models, there remains room for improvement. SCA is built on top of an LSTM backbone, which struggles with capturing long-range dependencies, limiting performance on complex weather sequences (sunny to rainy). Replacing LSTMs with Transformers could enhance the ability to model intricate temporal relationships while maintaining efficiency. Furthermore, the model may face challenges in generalizing to transitional weather conditions and handling long and intricate sequences. Future work could focus on incorporating domain adaptation and optimizing the architecture for longer input sequences.

3. **Unusual weather scenarios.** The dataset used to train VLAAD-TW includes a variety of transitional weather patterns to improve its performance in different conditions. These transitions, which have varying intensity levels over time, enable the model to better adapt to multiple scenarios. To further enhance the model's performance on previously unseen weather patterns, future work will explore using unsupervised domain adaptation.

4. **Transition image quality.** When transitioning between snowy and rainy conditions, image quality can suffer from distortion and noise due to the rain. To counteract this, future work will incorporate denoising algorithms as a preprocessing step. This will provide cleaner input images for the transition generation process, which should strengthen VLAAD-TW's ability to handle challenging weather shifts.

5. **Enhancing robustness to adversarial attacks.** High noise levels in the dataset can compromise model robustness, making it less reliable in real-world situations, especially with natural noise or adversarial perturbations. Future research will focus on developing strategies for both adversarial attacks and defenses to make VLAAD-TW more resilient once it is deployed.

6. **Physical plausibility.** While our framework effectively generates controllable visual transitions across diverse weather conditions, certain transformations (e.g., rain $\rightarrow$ snow) may show limited physical plausibility. Future work will aim to enhance the physical consistency of these transitions by incorporating physics-informed priors and temporal regularization to achieve smoother and more realistic weather dynamics.

7. **Risk-aware vision-language reasoning.** A promising and critical future direction for VLAAD-TW is the integration of real-time risk analysis into the VLM pipeline. While our current work provides semantic understanding of transitional weather scenarios, the next logical step is to move from description to proactive risk assessment. We propose extending our framework to localize potential hazards within the visual scene, such as slippery road conditions, obscured pedestrians, or hydroplaning risks. This would enable the model to generate not only descriptive captions but also actionable, risk-aware reasoning, such as "Caution: The road ahead is wet and a car is braking suddenly." Coupling risk localization with contextual explanations enables a more informed and interpretable basis for decision-making. Furthermore, incorporating causal reasoning into the risk assessment, to distinguish true danger signals from weather-related visual noise, is vital for enhancing the safety, generalization, and trustworthiness of autonomous systems operating in these challenging dynamic environments (Malla et al., 2023; Zhang et al., 2023b).

## 8   Conclusion

In this paper, we propose VLAAD-TW (Vision-Language Assistance for Autonomous Driving under Transitional Weather), a novel framework for image captioning and VQA in challenging weather conditions. VLAAD-TW introduces a novel cross-modal, spatiotemporal reasoning architecture specifically engineered for interpretable perception in autonomous driving under transitional weather conditions. It integrates a Feature Encoder for Transitional Weather (FETW), built on an efficient MobileNetV2 backbone, for robust visual feature extraction. A Spatiotemporal Contextual Aggregator (SCA), based on cascaded LSTM networks, processes input text to provide rich linguistic context. A Selective Attention-Guided Fusion Module (SAFM) then dynamically weighs and combines information from both modalities. Finally, a Semantic Text Generator (STG) produces precise, context-aware text from this fused representation. Our key contributions include a lightweight, computationally efficient architecture that significantly reduces model complexity compared to the baseline, achieving an 85% reduction in parameters, an 80% reduction in FLOPs and a 73% reduction in memory requirements while maintaining competitive performance. Also, we introduce AIWD16-text, a comprehensive vision-language dataset that features a rich collection of images spanning sixteen distinct transitional weather states, generated using a Stochastic Conditional VAE (SC-VAE). The images are further enriched with manually annotated image captions and open-ended question-answer pairs, making AIWD16-text a unique and valuable resource for training and evaluating vision-language models in this underexplored domain. Extensive experiments demonstrate VLAAD-TW's superior performance, achieving a 2.01% BLEU and 0.62% ROUGE-L improvement in image captioning and 0.6% and 0.83% gains

in VQA tasks while using significantly fewer parameters than existing methods. Notably, our model maintains robust performance across transitional weather. Future work will focus on data generation in dynamic backgrounds by integrating moving scene information alongside variations in weather intensity.

## 9 Acknowledgement

This work was supported in part by the NMICPS Technology Innovation Hub on Autonomous Navigation (Ti-HAN) at the Indian Institute of Technology Hyderabad under Project No. TiHAN-IITH/03/2024-25/037.

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
