# OpenReview forum: "Eyes on the Road, Words in the Changing Skies: Vision-Language Assistance for Autonomous Driving in Transitional Weather"
_TMLR — Accepted by TMLR_

### Review · Reviewer_fY8v · 2025-09-22

**Summary Of Contributions:**

To address these limitations that variable weather conditions for MLLM, this paper propose Vision-language Assistance for Autonomous Driving under Transitional Weather (VLAAD-TW). The VLAAD-TW integrates a Feature Encoder for Transitional Weather (FETW) for robust visual feature extraction, with a Spatiotemporal Contextual Aggregator (SCA), which models dynamic weather-induced changes. It also uses a Selective Attention-guided Fusion Module (SAFM) to balance visual and linguistic cues for a unified representation. Finally, a Semantic Text Generator (STG) fuses these representations to produce context-aware driving information. Furthermore, this paper introduce AIWD16-text dataset, an adverse intermediate weather driving dataset for vision language task.

**Audience:**

Yes

**Audience Explanation:**

The weather change may lead to a negative impact on vision language task for autonomous driving. And this paper provide a dataset and solution to address it.

**Claims And Evidence:**

Yes

**Claims Explanation:**

This paper provide experiments and dataset statistic to support the claims.

**Requested Changes:**

(1) In fact, there are many robust benchmark, including weather changes for autonomous driving (e.g., Robo3D [1]). Author should clarify the difference except for providing caption, which is easy to obtain like OmniDrive [2].

(2) I suggest authors providing more experiments about the VLAAD-TW trained on AIWD16-text. For example, first pretrain VLAAD-TW on AIWD16-text and then finetune it on nuScenes dataset and compare the performance with other methods (e.g., OmniDrive) to validate the effectiveness of VLAAD-TW and AIWD16-text.

(3) This paper adopt generation models to produce different weather conditions, which is not reliable and may harm the performance of MLLM on the realistic data.

[1] Robo3D: Towards Robust and Reliable 3D Perception against Corruptions

[2] OmniDrive: A Holistic Vision-Language Dataset for Autonomous Driving with Counterfactual Reasoning

---

### Review · Reviewer_g2A7 · 2025-10-10

**Summary Of Contributions:**

This paper tackles weather-transition modeling in AVT-VQA. Its key contributions are:
- SC-VAE, a data-synthesis framework that interpolates weather conditions from the Weather-Driving (WD) dataset.
- VLAAD-TW, an efficient yet high-performing VQA architecture.

**Audience:**

Yes

**Audience Explanation:**

1.	Highlights an under-explored data-augmentation angle (weather transitions) that deserves more attention.
2.	Delivers value on both the data (new benchmark) and model-design fronts.

**Claims And Evidence:**

Yes

**Claims Explanation:**

**Strengths**

Overall, the claims are well supported by sufficient experiements, e.g.,
1. The quality of the data are verified by Table 4 etc
2. The validity of the model is verified by Table 5~9

**Weaknesses**

1.	Significance of transitions. If extreme conditions (e.g., cloudy and foggy) are already available, how much does modeling the intermediate transition help? An ablation that compares VLAAD-TW trained on (a) WD alone vs. (b) AIWD16-text is essential to quantify the gain.
2.	Generalization of AIWD16-text. Demonstrate that models fine-tuned on the synthetic data do not degrade on realistic benchmarks; otherwise the utility of the new dataset is unclear.
3.	Scope beyond perception. Because the target domain is autonomous driving, include QAs that go beyond scene understanding (e.g., decision-making, planning) to broaden impact.

**Requested Changes:**

Besides those mentioned above in the weaknesses：

1.	PSNR/SNR evaluation. These metrics need ground-truth frames. Are only the first (t = 0) and last (t = T) frames used, or are intermediate frames also available? Clarify protocol.
2.	Transition realism. Some sequences (rain → snow) look unnatural; please comment on physical plausibility or provide user-study validation.
3.	Pre-training vs. fine-tuning. Specify which components of VLAAD-TW and the baselines (BLIP, etc.) are pre-trained, whether all models are fine-tuned on AIWD16-text, and whether DriveLM comparisons use identically tuned checkpoints. Missing details impede reproducibility.

Minor
Typo: “multi-model instruction tuning” → “multi-modal instruction tuning.” When a citation is used as a noun, drop the parentheses (e.g., “Park et al. 2024 introduced…”).

---

### Review · Reviewer_EgsA · 2025-11-14

**Summary Of Contributions:**

The paper leverage VLM for autonomous driving scenes perception in transitional weather. To improve the performance of VLM in autonomous driving tasks, the authors integartes a feature encoder for transitional weather, and a lightweight backbone for robust visual feature extraction. A fusion module is proposed to balance visual and linguistic cues, as well as a text generator. New dataset is contributed and the evaluations are performed on this dataset.

**Additional Comments:**

I'm not familiar with VLMs and its application in autonomous driving.

**Audience:**

Yes

**Audience Explanation:**

It is important in VLM for autonomous driving

**Claims And Evidence:**

Yes

**Claims Explanation:**

Extensive experiments are provided to support the authors claims

**Requested Changes:**

Please provide ablation of each proposed module in the pipeline, both qualitatively and quantitatively.

---

### Decision · Action_Editor_SrwU · 2025-12-24

**Recommendation:** Accept as is

**Audience:**

Yes

**Audience Explanation:**

Autonomous-Driving research community will benefit from this paper.

**Claims And Evidence:**

Yes

**Claims Explanation:**

This paper presents a well-motivated study on visual–language assistance for autonomous driving under transitional weather, supported by a new dataset and complementary model contributions. Reviewers find the problem important and agree that the proposed synthesis framework and the VLAAD-TW architecture are validated through extensive experimental evidence. The authors have satisfactorily addressed the major concerns raised, particularly regarding the significance of transitional conditions and dataset generalization, strengthening the overall contribution. Based on the positive assessments and the coherence of the empirical results, the paper is recommend for acceptance.